# Balanced Scaling Using Nonlinear Dynamic Metrics in Multivariate Time Series Modeling

## Abstract

Time-series foundation models have shown strong capability in tasks such as forecasting across diverse domains by leveraging informative waveform representations. The main challenge in building a generic multivariate time series model lies in adaptability and consistent pattern extraction across systems that differ in autocorrelation, sensitivity to initial conditions, and the complexity of their underlying dynamical structure, whether reflected in univariate or multivariate signals. Prior approaches often fall into two extremes: specialized models trained separately for individual systems or large-scale foundation models trained on heterogeneous collections of time series with limited dynamical grounding. Motivated by the Platonic Representation Hypothesis, we achieve a heuristic observation that models across domains tend to converge toward a shared representation space that encompasses systems expressible in time-series form, including systems governed by differential equations, canonical analytical functions, and stochastic processes. In this work, we introduce Pangu-TS, a Pre-trained modality Agnostic Network for Generic multivariate Time Series modeling. Pangu-TS is pre-trained on a benchmark dataset designed with a more balanced distribution of types of time series systems, quantified by several nonlinear dynamical metrics from chaotic theory. Through this analysis, we uncover an empirical *balancing law*, showing that maintaining representative distributions of dynamical systems is essential for controlling the patterns learned by the model. Alongside this, Pangu-TS demonstrates both strong zero-shot forecasting ability in real-world data and promising latent representation quality on various downstream tasks, as validated across benchmarks in fields of digital healthcare, battery life health, and civil monitoring.

## 1 Introduction and Background

Across scientific and engineering domains, time series serve as the primary record of how systems evolve over time. Extracting meaningful patterns, beyond statistical features such as the representation of different waveforms and temporal relationships (Luo et al., 2024), from these signals is essential for prediction, control, and scientific discovery. However, developing models that can robustly capture dynamics across heterogeneous multivariate systems is still an open problem. Building on this context, a number of prior works have laid important foundations for generic time series modeling. While these studies provide valuable insights, each exhibits certain limitations that leave open opportunities for further improvement. To contextualize our contribution, we summarize the literature below, highlighting its strengths, research gaps, and the motivations behind our approach.

**Time series forecasting.** Recent time series foundation models have shown strong out-of-the-box forecasting ability. Chronos (Ansari et al., 2024) reformulates forecasting as sequence modeling by quantizing values and training T5-style transformers, delivering competitive zero-shot probabilistic results on a 42-dataset benchmark. Sundial advances a generative, patch-based objective (TimeFlow Loss) that avoids discrete tokenization, pre-trains on the 1-trillion-point TimeBench, and reports state-of-the-art zero-shot results (Liu et al., 2025). Complementing these corpus-driven approaches, Panda pre-trains on a synthetic library of chaotic ODE systems discovered by evolutionary search and shows zero-shot transfer to real chaotic systems, emergent cross-channel structure, and even PDE generalization (Lai et al., 2025). Despite these advances, both Chronos and Sundial are not specifically tailored for multivariate forecasting; and their evaluations are dominated by operational/civil-monitoring datasets (e.g., energy, transport, retail) with heterogeneous protocols

and relatively limited coverage of mechanistic scientific systems, a trend also seen in several prior works (Das et al., 2024; Rasul et al., 2023; Woo et al., 2024).

**Multivariate time series modeling.** Multivariate time-series modeling remains without a unified framework or methodology, especially for generative forecasting tasks. For instance, CONVTRAN (Foumani et al., 2024) proposes a strong convolution–transformer hybrid with comprehensive evaluation across benchmarks, yet it is designed primarily for classification rather than pretraining or generation. CBRAMOD (Wang et al., 2025) introduces an adaptive multi-channel framework tailored for EEG analysis, showing robustness across subjects, but its scope is limited to neural signals and does not extend to general-purpose generative modeling. More recent approaches such as PANDA (Lai et al., 2025) and NORMWEAR (Luo et al., 2024) move toward broader applicability by introducing pre-trained multichannel models that claim generality across chaotic dynamical systems or arbitrary sensor signals, respectively. Their underlying philosophies are largely aligned, though with different application domains, also implicitly indicating the exist of Platonic representation hypothesis (Huh et al., 2024). Still, PANDA does not provide in-depth analysis of the representation quality learned by its model, while NORMWEAR leaves the generative potential of its pretraining framework underexplored. As a result, multivariate time-series modeling continues to lack a comprehensive pre-trained foundation that addresses both representation learning and generative forecasting in a unified manner.

**Benchmark Data.** After surveying existing benchmarks for multivariate time-series data, we identified the following representative datasets. SUNDIAL and related prior work (Liu et al., 2024) were originally evaluated on the benchmark introduced by Wu et al. (2021), which consists primarily of civil forecasting datasets such as electricity consumption, exchange rates, and weather. PANDA, in contrast, is assessed mainly on the chaotic systems benchmark released by Gilpin (2021). In the domain of materials science, Tan et al. (2025) recently introduced a battery life-cycle benchmark comprising profiles from batteries with varying materials and experimental configurations. Finally, wearable sensing represents another major corpus of time-series data, with Luo et al. (2024) releasing a benchmark that covers a wide range of commonly observed wearable signals, including those capturing physical activity, cardiovascular, and neural dynamics.

**Our Contribution.** To address the research gaps described above, we introduce PANGU-TS, a pre-trained modality agnostic network for generic multivariate time series modeling. PANGU-TS is pre-trained on a benchmark dataset designed with a more balanced distribution of system types, quantified by several nonlinear dynamical metrics from chaotic theory. We mainly leverage the pre-training benchmark released by Gilpin (2021) and Luo et al. (2024) for analyzing the balance of the time series with varied chaotic behavior and for pre-training, and the test data from civial monitoring (Wu et al., 2021), battery life cycle test (Tan et al., 2025), chaotic systems (Lai et al., 2025), and wearable signals (Luo et al., 2024) for evaluation on both generative tasks and downstream inference tasks. The model demonstrates both strong zero-shot forecasting ability in these real-world data and promising latent representation quality. Our key contributions are thus as follows:

- We introduce a chaotic theory based schema to inspect the balance of time series system distribution within a dataset, mainly focusing on evaluating extent of autocorrelation, sensitivity to initial condition, and topological complexity. Such a schema allow us to better understand the quality of the pre-training dataset, hence have a better control over the pattern learned by a pre-trained model.

- Motivated by the Platonic Representation Hypothesis, that pretrained models across heterogeneous domains tend to converge toward a shared latent space, we notice that it is important to consider the comprehensiveness of the pretrainng dataset. Such observation is originated mainly from our analysis on data balance, quantified through chaos-theoretic metrics, acts as a complementary axis to scaling, indicating that beyond simply enlarging datasets, maintaining balance is essential for improving representation quality.

- We pre-train a paradigm multivariate time series model for generic purpose on the integrated more chaotic-balanced dataset, with all the exploration and ablation studies conducted with data scale of $10^5$ number of samples, which contain approximately total time points around $10^9$. Our model demonstrate leading performance on 74 testing scenarios, comprising both generative based tasks and downstream inference tasks, in unseen real-world time series systems, including profiles of observed chaotic system, digital health data with tailored sensor configuration, battery life cycle test data with varied materials, and civil monitoring.

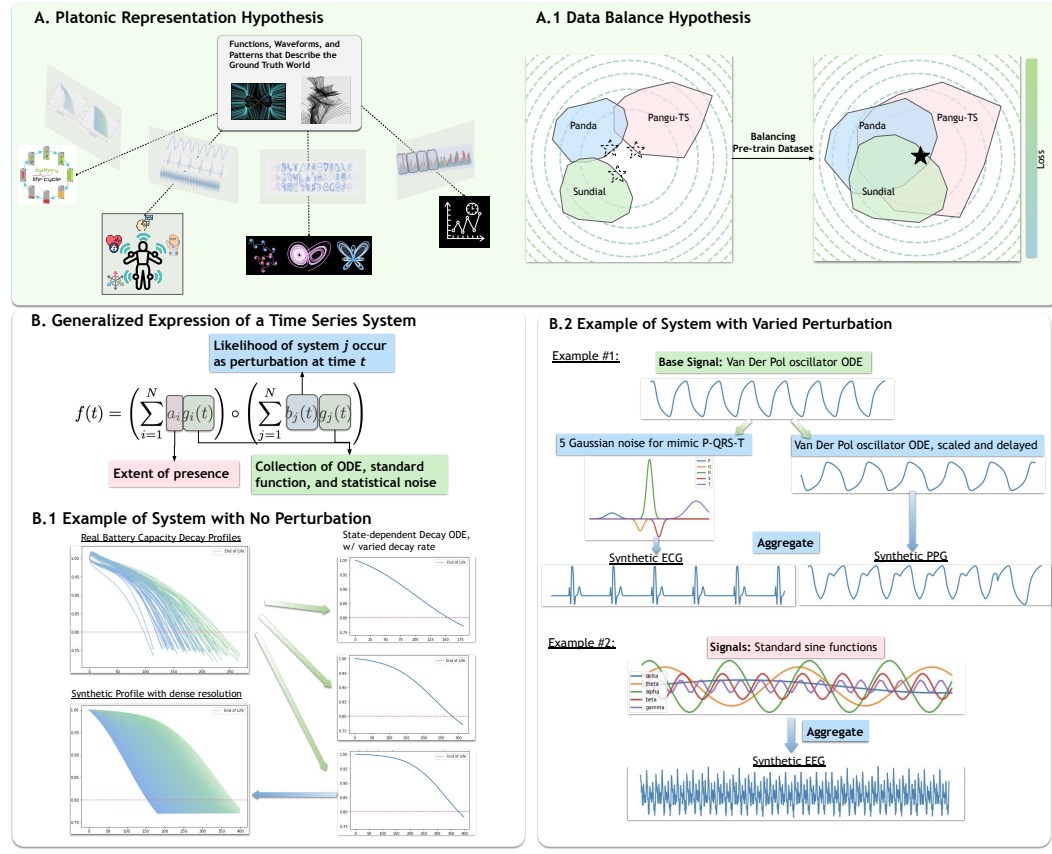

Figure 1: **Overview of the motivation.** (A) Demonstration of Platonic Representation Hypothesis in time series domain. (B) Heuristically generic expression for describing a observed time series system containing perturbation.

## 2 MOTIVATION FROM PLATONIC REPRESENTATION HYPOTHESIS

Huh et al. (2024) has claimed that models are converging in terms of the learned representation across varied modalities, model scales, and tasks of interest. In the field of time series, it can be depicted more explicitly. Specifically, time series data is essentially a set of points sampled with a pre-defined sampling rate, ranging from hundreds or thousands of Hz to one point per year.

For an arbitrary observed segment of time series, we know that when sampling rate goes to infinity, the continuous time series can be approximate by a Newton polynomial (Newton, 1833). However, Newton polynomial comprise limited information about the past unobserved data, as well as the indication toward future evolvement of the system. For real-world time series data like wearable sensing signals, object movement in space, and civil monitoring system that contain inner balancing mechanism and some extent of periodicity, Fourier series approximation (Freeman et al., 1878) turns out to be a better expression. Although under ideal scenario, Fourier series approximation can heuristically fit on any time series data, it is inefficient to dynamically adapting the real-world system which often contain stochastic process and perturbation of signals from varied sources. Thus, we achieve the following heuristic representation of time series:

$$f(t) = \left(\sum_{i=1}^{N} a_i g_i(t)\right) \circ \left(\sum_{j=1}^{N} b_j(t) g_j(t)\right) \tag{1}$$

where in $g$ is a collection of $N$ numbers of ODE, standard function such as linear, exponential, sine, and cosine, and statistical noise such as Gaussian noise; $a_i$ is the coefficient of extent of presence of the $i$th system, $\circ$ stands for an aggregation operation such as linear combination or max pooling, and $b_j(t)$ is the function representing the likelihood of $i$th system occur as a perturbation signal at time $t$. As a result, varied situation of systems are naturally presented in the current benchmark datasets:

- If a perturbation signal occur synchronously with the base signal, then it is equivalent to the aggregation mechanism leveraged by Lai et al. (2025) to achieve novel chaotic system.

- If a perturbation signal occur at a particular ongoing timestamp, and it eventually substitute the base signal, then it is equivalent to the Mixup mechanism leveraged in Luo et al. (2024) to achieve semi-synthetic wearable signals.

- If the perturbation occur with stochastic pattern and with some extent of decay factor due to unmeasurable contextual factors in real-world situation, then it is equivalent to the raw recorded time series data integrated in Luo et al. (2024); Liu et al. (2025) and the testing dataset in Lai et al. (2025).

- If several time series system presence and they perturbate each other with uneven factors, then it is equivelent to the data augmentation method leveraged in Ansari et al. (2024).

Some example of synthetic time series generated by logic of equation 1 is presented in Figure 1 (A.2 and A.3). Up to this point, we observe from a heuristic reasoning that all the time series system in recent representative modeling benchmarks are generated from the same essence that can be sufficiently expressed by Equation 1, consisting base signal(s) and perturbation signal(s) occurred at some timestamp due to contextual factors (some are measurable some are not) which is eventually presented as stochastic behaviors in those time series systems.

With this lens in place, the focus of our work turns to a more practical and consequential question: *how to evaluate the extent of perturbation, or in other words, the degree of chaotic behavior in the observed time series system?* We argue that assessing such a degree of chaos on a given pretrained dataset is crucial because it directly relates to pretraining data quality and, consequently, test-time performance. It is at this stage that our exploration of balancing law and methodological development naturally follows.

## 3 METHODOLOGY

### 3.1 EVALUATE DATA BALANCE WITH METRICS FROM CHAOTIC THEORY

Based on the Platonic representation hypothesis discussed above, we can further infer that the extent of perturbation in varied forms can be quantified by the extent of chaotic or predictability of an observed time series system. We then propose using a set of metrics from chaotic theory to evaluate how different types of chaos distributed within a collection of observed time series system. These metrics, following the guidance from literature in chaos theory, comprise detrended fluctuation analysis exponent (DFA) (Hu et al., 2001), Lyapunov exponent (LE) (Wolf et al., 1985; Kantz & Schreiber, 2003), and persistent entropy (PE) (Atienza et al., 2019; 2020) at zero and one dimension homology. These chaotic metrics mainly evaluate the extent of autocorrelation, sensitivity to initial condition, and connect and loop complexity in terms of the transformed topological structure of a given time series system respectively. Consequently, various dynamical system types are identified via a deterministic procedure that involves computing chaos metrics, performing unsupervised clustering, and categorizing cluster types according to fixed thresholds on each metric, following established practices in the literature (Hu et al., 2001; Kantz & Schreiber, 2003; Atienza et al., 2019).

Specifically, to inspect the data balance according to these chaotic metrics, we conduct K-Means clustering after calculating these metrics on each of the time series sample. The optimal number of clusters are identified based on Elbow rule. Detailed clustering and the associated interpretation of each cluster in terms of chaotic systems are presented in Appendix A and B. From Figure 2, we observed that both pretrained benchmark datasets of multivariate time series in Lai et al. (2025) and Luo et al. (2024) are dominate by one type of chaotic patterns. After aggregating these two multivariate benchmarks, we observed that different type of chaotic patterns are distributed with relatively more homogeneity, without readily apparent dominance from just one type of system as observed through both the statistical bar plot and the T-SNE dimension reduction visualization.

### 3.2 PRETRAINING AND INFERENCE

**Model backbone.** Regarding the backbone model leveraged in this study, we rely on the channel-aware mechanism for multivariate signal modeling proposed by Luo et al. (2024) with consideration based on computation efficiency. Detailed complexity analysis of multivariate time series modeling approaches (Luo et al., 2024; Lai et al., 2025; Liu et al., 2025) are presented in Appendix F. Detailed model structure is presented in Figure 3 section A, with the same encoding backbone as proposed by Luo et al. (2024), but optimized logic for initial time series patch embedding and the lightweight

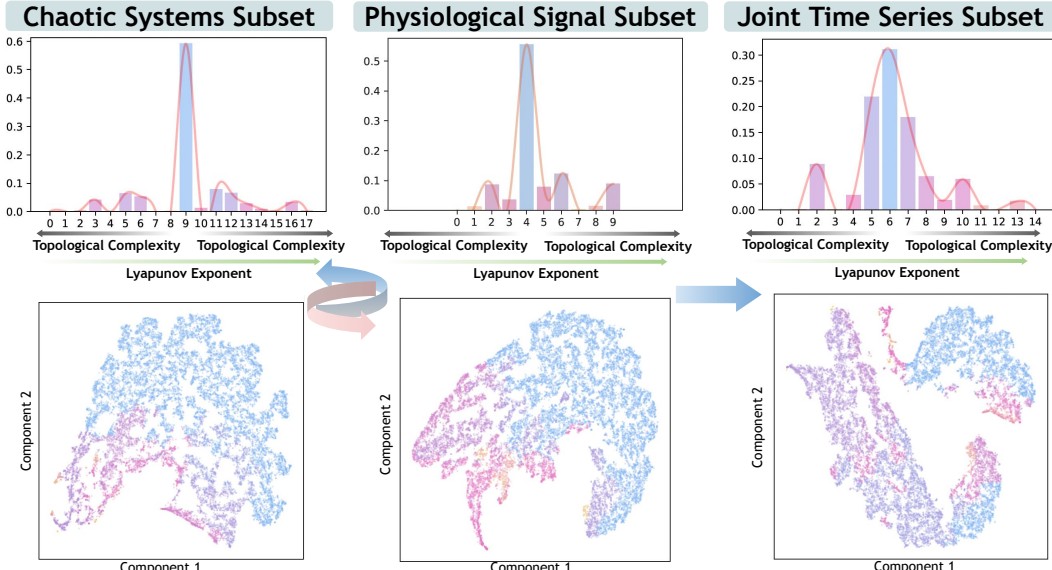

Figure 2: **Inspection of the balance using chao theory based metrics.** The histogram shows the presence density of different dynamical system types produced by the balance-inspection pipeline. For clarity, we order the bars by increasing Lyapunov exponent, placing systems with lower topological complexity farther from the center. We observe that both the chaotic-system subset and the physiological-signal subset are each dominated by a single system type. While in the joint time series subset, the distribution becomes more balanced, despite some distortion in the t-SNE view, with multiple system types exhibiting more comparable presence densities.

decoding block to better adapt the scenario for generic generative based multivariate time series modeling purpose.

**Training process.** The backbone model is pretrained on the aggregated pretraining data benchmark as discussed in Section 3.1, in a masking and reconstructing manner. After the input multivariate time series being patchified, the patches are randomly replaced with a trainable unified [MASK] token representation with a fixed probability threshold pre-defined following guidance from Huang et al. (2022). The masks applied on the input are independently sampled for each channel in the multivariate input, thus, varied masking combination are expected to be covered as more pretraining iterations progresses.

**Inference logic.** During inference time, we focus on two types of generative tasks in this study: forecasting and simulation. Under forecasting manner, the generation logic is equivalent to masking the latter part of a given time series. On the other hand, we define simulation task as completing the unobserved channel conditioned on one or more given channel. We refer this as simulation because it naturally align with varied application scenarios such as healthcare and battery testing that usually contain one or more controlled variables represented as separate input time series channels. An indication of the generation logic is presented in Figure 3 section C.

### 3.3 DOWNSTREAM EVALUATION

To evaluate the pre-trained multivariate time series model, we consider two main aspects: generation quality and representation quality. For the latter, we include several representative domains, namely chaotic systems (Lai et al., 2025), wearable sensing for digital health (Luo et al., 2024), battery degradation monitoring (Tan et al., 2025), and civil infrastructure monitoring (Wu et al., 2021), also as demonstrated in Figure 3 section B. These domains are chosen because they span a broad spectrum of dynamical regimes (from nonlinear chaotic dynamics to long-term degradation processes) and real-world applications (ranging from healthcare to engineering systems), thereby providing a comprehensive testbed for assessing both generalization and robustness.

Throughout evaluation across all tasks and models, we follow two key criteria. First, all testing samples are entirely unseen during pre-training; for instance, the chaotic systems come from real-world observations, healthcare applications use completely different sensor configurations from those in the pre-training dataset, and both battery and civil monitoring data are drawn from domains not

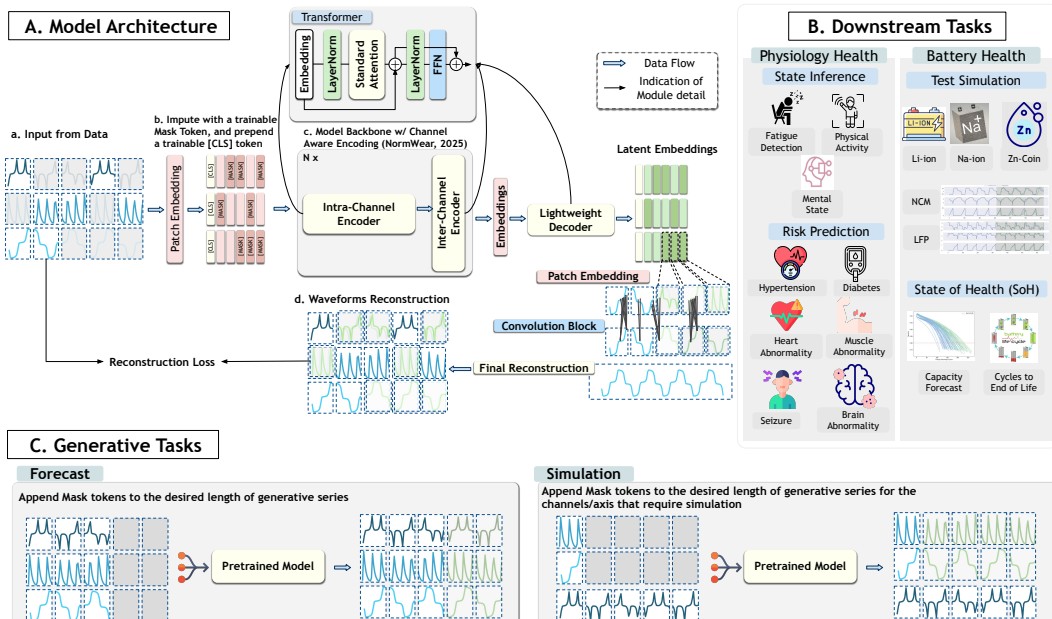

Figure 3: **Modeling schema.** (A) Detailed model architecture. (B) Domains of the downstream tasks for evaluating the representation quality of the pretrained multivariate time series models. (C) Inference logic during test-time for the two main generation based tasks focused in this study.

present during pre-training. Second, all evaluation data are multivariate with varying numbers of input channels, which is essential for testing and validating the robustness of the models.

When selecting models for comparison, we primarily focused on methods designed for multivariate time series modeling. Previous works (Lai et al., 2025; Luo et al., 2024; Liu et al., 2025) have comprehensively compared multivariate vs. univariate pretrained models (Ansari et al., 2024; Das et al., 2024; Woo et al., 2024; Shi et al., 2024), consistently showing that multivariate models outperform univariate ones across both generative tasks and downstream inference tasks. Therefore, in our evaluation we focus on multivariate pretrained baselines (Lai et al., 2025; Liu et al., 2025), which represent the state-of-the-art for generic multivariate time series modeling approach.

## 4 EXPERIMENTAL RESULTS

### 4.1 EVALUATION ON GENERATIVE AND DOWNSTREAM TASKS

**Setup.** We evaluate the proposed model along two dimensions: (i) *generation quality*, measured by mean absolute error (MAE) and mean squared error (MSE), and (ii) *representation quality*, assessed through downstream classification and regression tasks. For a fair comparison, all models observe the first 50% of each test sequence. Short-term and long-term forecasting correspond to generating 50% and 100% of the unobserved segments, respectively. In simulation tasks, we follow domain-specific configurations: for example, in battery health data the current channel is controlled and models are required to generate voltage and capacity given current inputs, while for other multivariate datasets we apply random channel masking (67%) following Narayanswamy et al. (2024). For downstream evaluation, we adopt AUC-ROC for healthcare classification, mean relative accuracy for healthcare regression (Luo et al., 2024), and mean absolute percentage error (MAPE) for battery state-of-health (SoH) prediction (Tan et al., 2025).

**Overall performance.** Across forecasting, simulation, classification, and regression, PANGU-TS achieves the strongest overall results. Figure 4 provides a visual summary, with quantitative scores reported in Table 6. Compared with PANDA, which is trained primarily on a single type of perturbation, PANGU-TS benefits from exposure to a more balanced dataset, resulting in higher-quality multivariate time series generation. In addition, PANGU-TS yields superior latent representations, as evidenced by consistently better downstream performance on both digital health tasks with varied sensor configurations and battery health tasks with heterogeneous material compositions.

**Detailed observations.** We further note several domain-specific trends. PANGU-TS trained on chaotic and sensor-rich data shows strong gains in wearable sensing and battery health tasks, while

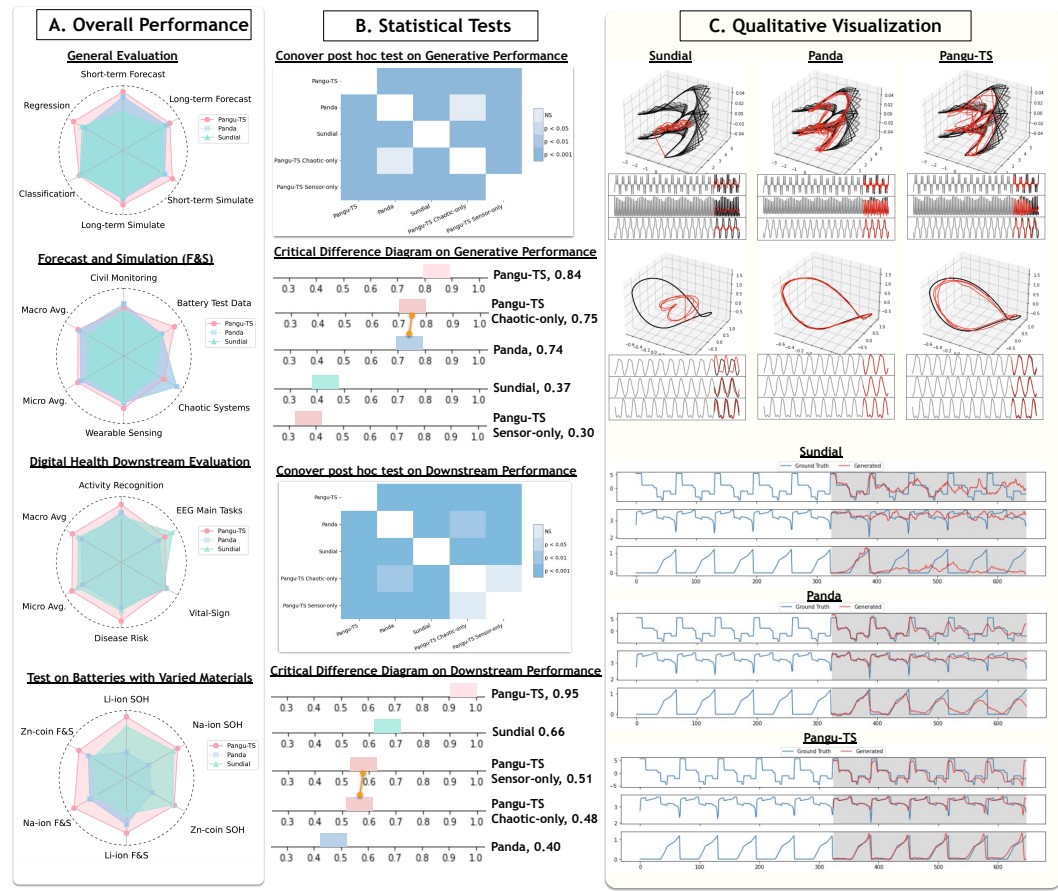

Figure 4: **Overview of the evaluation results.** (A) Radar plots showing the overall performance when evaluating the pre-trained models from varied aspects. (B) Statistical test accompanied with a critical difference diagram (with crossbar indicating no statistical significance) on the model's performance across all the domains and tasks (Demšar, 2006). (C) Examples for visualizing the generative quality of the models during test time.

PANDA retains relative advantage on chaotic and civil monitoring systems. Both models outperform SUNDIAL across domains. When trained only on chaotic data, PANGU-TS exhibits behavior closer to PANDA, highlighting the importance of balanced training. In terms of efficiency, PANGU-TS and PANDA achieve comparable inference speeds and are substantially faster than SUNDIAL.

**Statistical significance.** To verify the robustness of performance differences, we conduct statistical significance tests on model rankings using the Conover post-hoc test and Critical Difference Diagrams (Demšar, 2006; Pillai et al., 2024; Luo et al., 2024). Results in Figure 4 panel B confirm that the observed improvements of PANGU-TS over baselines are statistically significant.

## 4.2 ABLATION STUDIES: OBSERVATION OF BALANCING LAW

Previous studies have extensively demonstrated the existence of scaling laws in signal modeling. In this work, we posit that the balancing law serves as an accompanying factor that should be considered jointly with scaling. We present the ablation study results illustrating the balancing law. The chaotic-based balance score, as discussed, quantifies the distributional diversity of the pretraining dataset, capturing the balance between different types of dynamical systems, which we hypothesize is crucial for effective generative performance.

To examine the direct effect of balance, we construct subsets from the integrated benchmark dataset that share an identical level of data size of $10^5$ but differ in their balance scores. Balance scores are computed using weighted Shannon entropy and granularity-based diversity measures derived from clustering the underlying dynamical systems. As shown in Figure 5, models pretrained on datasets with higher balance consistently achieve lower generative error across multiple downstream tasks. These results indicate that balance alone, without changing data size, architecture, and training dynamics, correlates with improved generative behavior.

Table 1: Mean Absolute Error (MAE) for different methods across generative series. Values are reported as mean $\pm$ variance across the performance metrics from 4 sub-domains with generative tasks, 2 sub-domains with downstream tasks, and a total of 74 testing scenarios. The best performance are highlighted in bold, with second place being underlined. More detailed performance report are presented in Appendix H. Detailed information of the testing scenarios are specified in Appendix C.1.

| Models | SUNDIAL$_{large}$ | PANDA | PANGU-TS (Ours) | PANGU-TS Chaotic set | PANGU-TS Sensor only |
|---|---|---|---|---|---|
| **Metric** *(Zero-shot Generative Tasks)* | MAE ↓ | MAE ↓ | MAE ↓ | MAE ↓ | MAE ↓ |
| Short term forecast | 0.777 ± 0.042 | **0.540 ± 0.028** | 0.558 ± 0.010 | 0.595 ± 0.015 | 0.774 ± 0.042 |
| Long term forecast | 0.826 ± 0.034 | **0.661 ± 0.032** | 0.696 ± 0.019 | 0.730 ± 0.028 | 0.861 ± 0.040 |
| Short term simulate | 0.737 ± 0.024 | 0.718 ± 0.087 | **0.596 ± 0.011** | 0.632 ± 0.015 | 0.807 ± 0.039 |
| Long term simulate | 0.793 ± 0.022 | 0.770 ± 0.039 | **0.702 ± 0.018** | 0.732 ± 0.022 | 0.870 ± 0.034 |
| All short term generative | 0.757 ± 0.032 | 0.629 ± 0.063 | **0.577 ± 0.010** | 0.613 ± 0.015 | 0.790 ± 0.039 |
| All long term generative | 0.810 ± 0.027 | 0.715 ± 0.036 | **0.699 ± 0.017** | 0.731 ± 0.024 | 0.865 ± 0.035 |
| All generative series | 0.783 ± 0.029 | 0.672 ± 0.051 | **0.638 ± 0.017** | 0.672 ± 0.022 | 0.828 ± 0.038 |
| **Metric** *(Battery SOH Downstream)* | $R^2$ ↑ | $R^2$ ↑ | $R^2$ ↑ | $R^2$ ↑ | $R^2$ ↑ |
| Li-ion SOH | 0.040 ± 0.092 | -0.726 ± 0.010 | **0.229 ± 0.021** | -0.342 ± 0.007 | 0.084 ± 0.070 |
| Na-ion SOH | 0.954 ± 0.0001 | 0.879 ± 0.010 | **0.963 ± 0.0002** | 0.888 ± 0.002 | 0.947 ± 0.001 |
| Zn-coin SOH | 0.551 ± 0.046 | 0.255 ± 0.043 | **0.563 ± 0.012** | 0.447 ± 0.014 | 0.371 ± 0.088 |
| Battery Downstream Avg. | 0.500 ± 0.214 | 0.073 ± 0.884 | **0.531 ± 0.238** | 0.290 ± 0.243 | 0.328 ± 0.213 |
| **Metric** *(Wearable Downstream Tasks)* | AUC-ROC ↑ | AUC-ROC↑ | AUC-ROC ↑ | AUC-ROC ↑ | AUC-ROC ↑ |
| State Recognition | 0.790 ± 0.023 | 0.797 ± 0.025 | **0.813 ± 0.021** | 0.799 ± 0.025 | 0.790 ± 0.028 |
| EEG Tasks | **0.888 ± 0.026** | 0.856 ± 0.024 | 0.872 ± 0.027 | 0.862 ± 0.024 | 0.864 ± 0.026 |
| Vital Sign (1-MAPE) | 0.948 ± 0.002 | **0.954 ± 0.001** | 0.953 ± 0.001 | 0.949 ± 0.001 | 0.953 ± 0.001 |
| Disease Risk | 0.720 ± 0.047 | 0.709 ± 0.051 | **0.759 ± 0.034** | 0.741 ± 0.039 | 0.694 ± 0.053 |
| Wearable Downstream (class) Avg. | 0.801 ± 0.035 | 0.785 ± 0.035 | **0.815 ± 0.027** | 0.801 ± 0.029 | 0.782 ± 0.038 |
| 1$^{st}$ **Count** | 12 | 23 | **30** | 5 | 4 |

Table 2: Ablation Study: Performance comparison across models pretrained using different backbone architectures.

| Models | Chronos$_{base}$ (Ansari et al., 2024) | Uni-variate (Nie et al., 2023) | [CLS] Attn. (Luo et al., 2024) | Channel Attn. (Lai et al., 2025) |
|---|---|---|---|---|
| *Zero-shot Generative Tasks* | MAE ↓ | MAE ↓ | MAE ↓ | MAE ↓ |
| Short-term forecast | **0.433±0.018** | 0.564±0.019 | 0.558±0.010 | 0.653±0.032 |
| Long-term forecast | **0.638±0.083** | 0.694±0.022 | 0.696±0.019 | 0.747±0.030 |
| Short-term simulate | 0.451±0.018 | 0.569±0.012 | 0.596±0.011 | **0.405±0.053** |
| Long-term simulate | 0.696±0.135 | 0.706±0.019 | 0.702±0.018 | **0.477±0.062** |
| All short-term generative | **0.442±0.017** | 0.566±0.015 | 0.577±0.010 | 0.529±0.057 |
| All long-term generative | 0.667±0.105 | 0.700±0.020 | 0.699±0.017 | **0.612±0.063** |
| All generative series | **0.554±0.072** | 0.633±0.021 | 0.638±0.017 | 0.571±0.060 |
| *Battery SOH Downstream* | $R^2$ ↑ | $R^2$ ↑ | $R^2$ ↑ | $R^2$ ↑ |
| Li-ion SOH | 0.036±0.026 | -0.190±0.059 | **0.229±0.021** | -0.487±0.075 |
| Na-ion SOH | **0.966±0.000** | 0.525±0.255 | 0.963±0.0002 | -1.628±3.145 |
| Zn-coin SOH | 0.433±0.017 | **0.630±0.016** | 0.563±0.012 | 0.541±0.054 |
| Battery SOH Avg | 0.507±0.173 | 0.297±0.189 | **0.531±0.238** | -0.955±4.986 |
| *Wearable Downstream Tasks* | AUC-ROC ↑ | AUC-ROC↑ | AUC-ROC ↑ | AUC-ROC ↑ |
| State Recognition | 0.799±0.11 | 0.804±0.025 | 0.813±0.021 | **0.814±0.023** |
| EEG Tasks | 0.807±0.021 | **0.875±0.027** | 0.872±0.027 | 0.872±0.026 |
| Vital Sign (1-MAPE) | **0.953±0.001** | 0.951±0.001 | **0.953±0.001** | 0.949±0.001 |
| Disease Risk | 0.621±0.034 | 0.743±0.037 | **0.759±0.034** | 0.724±0.045 |
| Wearable Avg. | 0.768±0.032 | 0.832±0.028 | **0.838±0.025** | 0.826±0.031 |
| **Avg. Rank** | 2.250 | 2.688 | **2.188** | 2.750 |

In order to further strengthen the causal interpretation, we compare two controlled pretraining settings that manipulate balance scores and data size in opposite directions: a relatively small but balanced dataset versus a relatively large but unbalanced one. The balanced dataset, consisting of $1 \times 10^5$ samples with a balance score of 0.73, contrasts with the unbalanced dataset consisting of

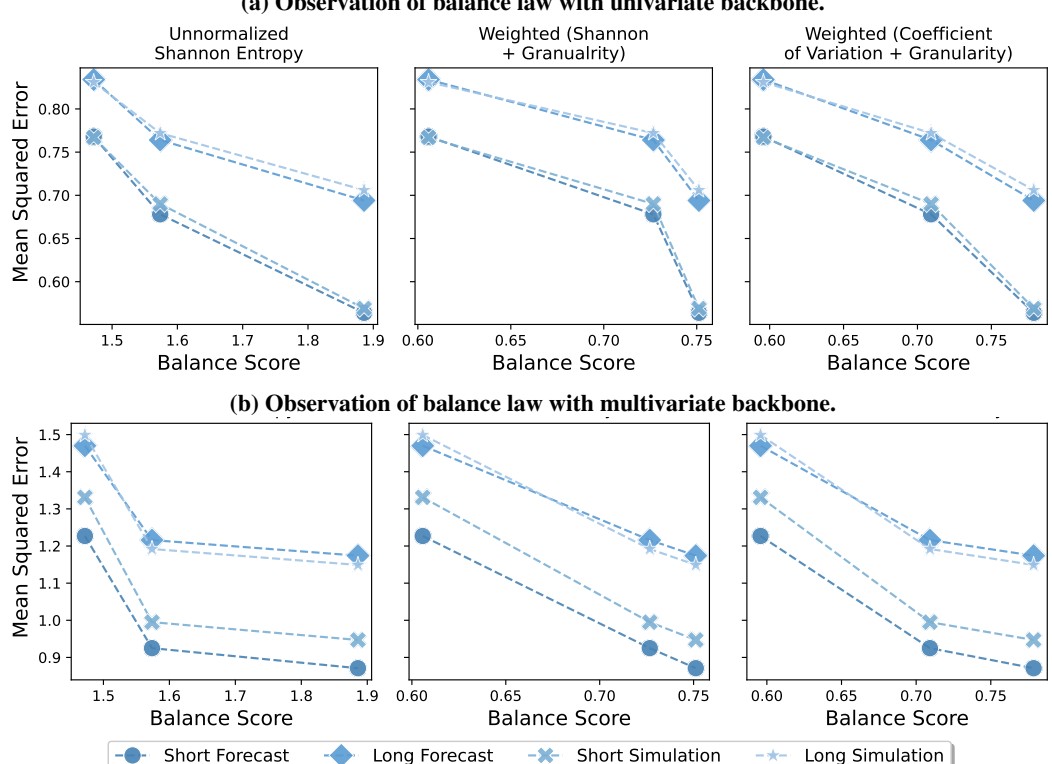

Figure 5: **Observation of Balancing Law.** We show performance on generative tasks across models pre-trained on datasets with varied balance score. The metrics leveraged for evaluation of data balance is stated at the top of each plot. Details of the metrics are presented in Appendix G. Results indicate that as pre-trained data becomes more balance in terms of covering more type of perturbations on time series, there are significant gains in the generative quality.

$2 \times 10^5$ samples with a balance score of 0.60. Balance scores are computed using weighted Shannon entropy and granularity measures derived from clustering distributions of different dynamical systems. All model checkpoints share identical architectures, hyperparameters, and pretraining procedures. As shown in the left panel of Figure 6, models pretrained on the smaller but more balanced dataset consistently outperform those trained on the larger, unbalanced dataset across various domains and test settings. This experiment rules out the possibility that the improvement observed in Figure 5 is simply a result of larger or richer datasets. Instead, it highlights the role of balance as an independent and influential dataset property.

Finally, to disentangle the relationship between scaling and balance, we conduct a third ablation study replicating the scaling-law pattern across datasets with different balance levels (balance scores = 0.60, 0.73, 0.75). The model architecture is fixed, while data size is varied. As illustrated in the right panel of Figure 6, scaling on more balanced datasets leads to a faster drop in test-time generative error, indicating that balance quality accelerates the benefit of scaling. For clarity of presentation, we did not include the results from the physiological signal subset in Figure 6, as its error scale is substantially higher than the other datasets. Nonetheless, these detailed results, reported in the Appendix J, are consistent with and further reinforce the observed trend.

## 5 CONCLUSION AND DISCUSSION

**Conclusion.** We presented PANGU-TS, a pre-trained foundation model for multivariate time series, alongside a chaos-theory-based framework for analyzing dataset composition.. Through extensive evaluation across forecasting, simulation, and downstream tasks, we demonstrated that PANGU-TS not only improves generative performance but also learns latent representations of high quality, enabling broad applicability across domains such as chaotic systems, digital health, battery monitoring, and civil infrastructure. Complementing the model itself, our analysis of dataset balance using chaos-theoretic metrics revealed an empirical *balancing law*, showing that maintaining a representa-

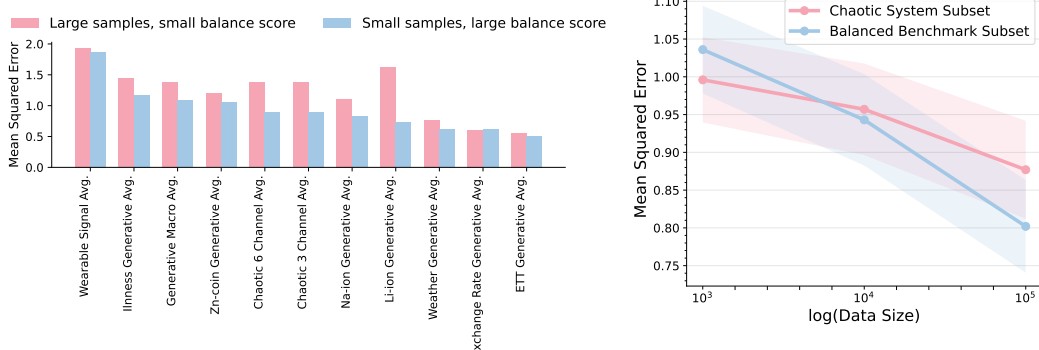

Figure 6: **Larger Data $\neq$ More Balance Data. Left:** We observed that model trained on relatively large but unbalanced subset didn't bring direct benefit in the representation quality. **Right:** We conduct ablation studies on scaling law, demonstrating that scaling with balanced presence of multivariate time series systems could provide more benefit in the performance. The average and standard deviation of mean performance across 44 generative tasks are reported.

tive distribution of dynamical systems in the pretraining data is essential for controlling the patterns captured by the model. This finding, alongside scaling considerations, underscores a dual axis for improving representation quality and supports the potential existence of Platonic Representation Hypothesis in domain of time series modeling.

**Limitation.** Firstly, as this paper is not intended as a theory contribution, the Platonic Representation Hypothesis introduced here remains at a conceptual level. Its rigorous validation from a theoretical standpoint will require contributions from theory-focused researchers, who can provide the formal underpinnings necessary to strengthen and generalize the hypothesis from theoretical perspective. Secondly, balancing the pretraining datasets introduces an inherent tradeoff: while it improves consistency and generality across diverse dynamical regimes, it can reduce performance in domains that rely heavily on specialized dynamics, such as chaotic systems. This reflects a structural limitation of the balancing process rather than a failure of the model itself. Understanding how such balancing reshapes the boundary between specialization and generality remains an open question and an important direction for future work.

**Broader Impact.** Our study highlights the potential of time series foundation models as building blocks for broader world modeling, offering unified representations that connect physical simulations, sensor data, and real-world observations. Apart from improved accuracy as extensively justified through this study, chaos based balance inspection essentially provides a principled and interpretable way to assess and control the diversity of dynamical systems in pretraining datasets. It offers insights into dataset quality, and complements scaling by ensuring the model captures representative patterns rather than being dominated by over-represented dynamics. Such insights can contribute to practical applications like digital twins, supporting reliable simulation and forecasting in domains from health monitoring to energy systems, while also serving as transferable tools that accelerate scientific discovery in areas such as materials science and climate modeling, where both comprehensive representations and faithful simulation of underlying dynamics are critical.

## ETHICS STATEMENT

This study contains applications in the field of healthcare. We ensured that all the data being used during pretraining and evaluations were made publicly available by the original authors, and all these works were cited properly.

## REPRODUCIBILITY STATEMENT

The full code base, comprising all the scripts for exploratory data analysis and preprocessing, model construction, pretraining, downstream evaluation, result analysis, and all the visualizations that are described in this paper, will be published simultaneously with the paper.

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

## A  PROCEDURE OF INSPECT EXTENT OF BALANCE USING CHAOS THEORY BASED METRICS

The balance inspection involve 3 main stages: (I) metrics computation, (II) unsupervised clustering, and (III) balance score computation. The first stage generally follows the procedure described in algorithm 1, which is implemented with multi-process manner as demonstrated in the codebase. The second stage follows the entirely automated pipeline described in algorithm 2, where the outcome is visualized in section B. Finally, the balance score is computed based on the outcome from the previous two stages, where the scoring details is presented in section G.

---

**Algorithm 1:** Compute Nonlinear Dynamics Metrics

---

**Input:** Pretraining dataset $\mathcal{D}$ of multichannel time series
**Output:** NLD metrics per channel: DFA, Persistence Entropy, Lyapunov Exponent

**foreach** *sample $X \in \mathcal{D}$* **do**

    **foreach** *channel $x$ in $X$* **do**

        **DFA:** $d \leftarrow \text{DFA}(x)$;

        **Persistence Entropy:**
            $\tilde{x} \leftarrow \text{TakensEmbed}(x;\ \text{delay} = 1, \dim = 5)$;
            $D \leftarrow \text{VietorisRipsPersistenceDiagram}(\tilde{x})$;
            $p \leftarrow \text{PersistenceEntropy}(D)$;

        **Lyapunov Exponent:**
            $\lambda \leftarrow \text{LyapunovExponent}(x;\ \text{embdim} = 10, \tau = 1, \text{minsep} = 10)$;

        Store $(d, p, \lambda)$;

---

---

**Algorithm 2:** Clustering and Dynamical-System Typing Based on NLD Metrics

---

**Input:** NLD feature matrix $\mathbf{F} \in \mathbb{R}^{N \times M}$
**Output:** Cluster types and merged histogram categories

**Step 1: Determine optimal cluster number.**
Compute K-means for $k = 2, 3, \ldots, K_{\max}$;
Record final inertia value for each $k$;
Select $k^\star >> k_{optimal}$ where $k_{optmial}$ is identified using the elbow rule.

**Step 2: Clustering**
Fit K-means with $k^\star$ and obtain centroids $\{c_1, \ldots, c_{k^\star}\}$.

**Step 3: Assign semantic labels to centroids.**
Compute global feature means $\mu$ across all centroids.
**for** $j = 1$ **to** $k^\star$ **do**
  Initialize label string $L_j$;

  `// DFA: correlation or stationarity`
  **if** $c_j[\text{DFA}] < 0.5$ **then**
  │  append "Anti-corr" to $L_j$;
  **else**
  │  **if** $c_j[\text{DFA}] < 1.0$ **then**
  │  │  append "Positive-corr" to $L_j$;
  │  **else**
  │  └  append "Non-station" to $L_j$;

  `// Lyapunov exponent: degree of chaos`
  **if** $c_j[\lambda] < 0$ **then**
  │  append "Stable" to $L_j$;
  **else**
  │  **if** $c_j[\lambda] < \mu[\lambda]$ **then**
  │  │  append "Rel Chaos" to $L_j$;
  │  **else**
  │  └  append "Rel Very Chaos" to $L_j$;

  `// Persistent entropy: topological complexity`
  **if** $c_j[\text{PE}_{H0}] < \mu[\text{PE}_{H0}]$ **then**
  │  append "Low Connect Complex" to $L_j$;
  **else**
  └  append "High Connect Complex" to $L_j$;
  **if** $c_j[\text{PE}_{H1}] < \mu[\text{PE}_{H1}]$ **then**
  │  append "Low Loop Complex" to $L_j$;
  **else**
  └  append "High Loop Complex" to $L_j$;

  Assign $L_j$ as the type of centroid $c_j$;

**Step 4: Merge clusters with identical type labels.**
Group centroids sharing the same label and compute histogram.
**return** *merged cluster types and histogram*

---

# B CLUSTERING OF TIME SERIES SYSTEMS WITH CHAOTIC METRICS

Figure 7: **T-SNE plot of the datasets, quantified by the proposed metrics from chaotic theory**. This figure mainly specified the exact group of time series with different chaotic attributes corresponding to plot presented in Figure 1 Section B.

## C DATASETS

We leverage the datasets released by Luo et al. (2024) for 18 wearable downstream tasks, Lai et al. (2025)'s evaluation set for evaluating the generative quality on real-world chaotic system, Tan et al. (2025)'s datasets for the evaluation on battery test time series, and Wu et al. (2021)'s datasets for evaluation on civil monitoring forecasting tasks. The pre-train datasets are completely disjoint collections of multivariate time series data, with statistics presented in Table 3.

Table 3: **Pretrain Datasets.**

| Datasets | Sequence Length | # Samples | # Variates |
|---|---|---|---|
| Luo et al. (2024) | 390 | $2.3 \times 10^5$ | {2,3,4,6} |
| Lai et al. (2025) | 4096 | $1.0 \times 10^5$ | {3,4,6} |
| Aggregated Benchmark | {390, 4096} | $2.2 \times 10^5$ | {2,3,4,6} |

### C.1 TEST SCENARIOS INFORMATION

The detailed statistics of the downstream tasks including the battery state of health (SOH) prediction and digital health downstream tasks were presented in prior works in the literature: Tan et al. (2025) and Luo et al. (2024) respectively. The data statistics of the genrative tasks are summarized in Table 4 as shown below.

Table 4: Statistics of generative testing datasets used in our experiments.

| Dataset | Data Points per Channel | Channels | Domain |
|---|---|---|---|
| Panda Test Set (Lai et al., 2025) | $3.4 \times 10^7$ | [3, 6] | Chaotic System |
| WESAD (Luo et al., 2024) | $4.2 \times 10^6$ | [6] | Wearable Sensing |
| HUST (Tan et al., 2025) | $1.0 \times 10^6$ | [3] | Battery Health and Material Test |
| CALB (Tan et al., 2025) | $1.6 \times 10^4$ | [3] | Battery Health and Material Test |
| Na-ion (Tan et al., 2025) | $2.0 \times 10^4$ | [3] | Battery Health and Material Test |
| Zn-coin (Tan et al., 2025) | $6.4 \times 10^5$ | [3] | Battery Health and Material Test |
| ETT (Liu et al., 2025) | $1.7 \times 10^5$ | [7] | Civil Monitoring |
| Weather (Liu et al., 2025) | $5.2 \times 10^4$ | [21] | Civil Monitoring |
| illness (Liu et al., 2025) | $1.0 \times 10^3$ | [7] | Civil Monitoring |
| Exchange Rate (Liu et al., 2025) | $7.2 \times 10^3$ | [8] | Civil Monitoring |

## D MODEL IMPLEMENTATION DETAIL

We provide a formal description of the masking, encoding, and reconstruction pipeline used in PANGU-TS. Let the input multivariate time series be $X \in \mathbb{R}^{C \times T}$ where $C$ denotes the number of channels and $T$ the sequence length.

### D.1 PATCHIFICATION

Each channel is divided into non-overlapping patches of length $P$:

$$X^{(c)} \rightarrow \left\{ x_1^{(c)}, x_2^{(c)}, \ldots, x_L^{(c)} \right\}, \qquad x_i^{(c)} \in \mathbb{R}^P, \tag{2}$$

where $L = T/P$. Each patch is projected to a $D$-dimensional embedding through a Conv1D patch embedding module $E(\cdot)$:

$$z_i^{(c)} = E\left( x_i^{(c)} \right) \in \mathbb{R}^D. \tag{3}$$

where $D$ is the latent dimension.

### D.2 CHANNEL-WISE MASK SAMPLING

Masks are then sampled independently for each channel and each patch:

$$m_i^{(c)} \sim \text{Bernoulli}(p_{\text{mask}}), \tag{4}$$

where $m_i^{(c)}$ is the result of the indicator function with success or failure outcome sampled from the Bernoulli distribution. The masked embedding is then formed as

$$\tilde{z}_i^{(c)} = \left(1 - m_i^{(c)}\right) \cdot z_i^{(c)} + m_i^{(c)} \cdot E\left([\text{MASK}]\right). \tag{5}$$

where [MASK] is a single trainable vector similar to the [CLS] special token. This produces patch-level masked embeddings for all channels. Then these embeddings are sending to the core backbone as described in method section 3.2. Denote the output from the backbone as $\hat{z}_{i,latent}^{(c)} \in \mathbb{R}^D$.

### D.3 PATCH-LEVEL DECONVOLUTION

The latent embeddings generated from the backbone are then projected back to waveform patches through a lightweight DeConv1D module:

$$\hat{s}_i^{(c)} = \text{DeConv1D}\left(\hat{z}_{i,latent}^{(c)}\right) \in \mathbb{R}^{P \times K} \tag{6}$$

Such a deconvolve step generates multiple candidate reconstructions $\{\hat{s}_i^{(c,k)}\}_{k=1}^K$ where $K$ is the maximum number of candidates. These candidate reconstructions are then consolidated via a Conv1D module:

$$\hat{s}_i^{(c)} = \text{Conv1D}\left(\hat{s}_i^{(c,1)}, \ldots, \hat{s}_i^{(c,K)}\right) \in \mathbb{R}^{P \times 1} \tag{7}$$

and collectively, the final reconstruction is:

$$\hat{X_{\text{reconstruct}}} = \text{ConcatReshape}(\hat{s}_i^{(c)}, \ \forall c \in C, i \in L) \in \mathbb{R}^{C \times T}. \tag{8}$$

## E MODEL AND TRAINING CONFIGURATION

PANGU-TS is derived from the Masked Autoencoder (MAE) (He et al., 2021). The detailed hyper-parameter choice is describe in 5. We use a Conv1D layer with a kernel size of 16 and a stride of 16, ensuring no overlapping patches. This layer takes input with 1 channels and projects it to 768 channels, matching the hidden size of our encoders. In PANGU-TS, we apply random masking independently to each variate along both the frequency and time axes, with respective masking ratios of 0.5. The masked patch are replaced by a trainable mask special token before passing to the encoder. To enhance representation learning, following Luo et al. (2024), we introduce six additional transformer blocks as fusion layers, interleaved with the original 12 encoder blocks, creating a total of 18 blocks. Each transformer block has a hidden dimension of 768 and uses LayerNorm as in the original MAE. The latent embeddings obtained from the encoder are projected from 768 to 512 dimensions before passing to the decoding blocks. The positional embeddings are added to guide the decoder in reconstructing the input series. The lightweight decoder consists of two transformer blocks with a hidden dimension of 512, followed by two Conv1D layers. The first Conv1D layer maps from the flattened multivariate signal embedding to an intermediate dimension, and the second Conv1D layer maps from this intermediate dimension back to the original multivariate signal space. A GELU activation function is used between these layers, with BatchNorm applied to the input. The decoder reconstructs the original input series, and the model is trained using Mean Squared Error (MSE) loss on all data points. We did not perform on-the-fly data augmentation, as suggested in the MAE framework, due to the high masking ratio. (An end-to-end example of the input and output of this pretraining pipeline is illustrated in Fig. 3)

The models are pretrained on 4 NVIDIA RTX 3090 graphical computing unit (GPU), with 24GB of GPU memory on each card.

Table 5: **Pretraining Hyper-parameters.**

| Hyper-parameter | Value |
|---|---|
| # cross-patches Transformer Encoder | 12 |
| # cross-channels Transformer Encoder | 6 |
| # Transformer Decoder | 2 |
| # Attention Heads | 12 |
| Encoder Latent Size | 768 |
| Decoder Latent Size | 512 |
| Feedforward Latent Size | 3072 |
| Normalization | LayerNorm |
| Patch size | 9 |
| Optimizer | AdamW |
| Loss Scalar | NativeScaler |
| Base Learning Rate (blr) | 1e-3 |
| Epochs | 100 |
| Batch size | 128 |

## F    COMPLEXITY ANALYSIS OF VARIED CHANNEL-AWARE ENCODING MECHANISM

Among the three core pre-trained model comprised in this study, they leverage similar idea but different implementation to approach the multivariate time series modeling. SUNDIAL (Liu et al., 2025) propose *single-series sequence* (S3) for which they aggregate all the input series channels into a single channel along the temporal axis. When passing the time series in S3 format to the backbone model, the attention-based encoding schema is equivalent to the *All-Attention* mechanism discussed in Luo et al. (2024), which, as the authors stated, have the complexity of

$$M_{Sundial} = O(d \cdot (L \cdot C)^2) \tag{9}$$

where $d$ is the embedding size, $L$ as the sequence length, and $C$ is the number of input variate or series channel. Such running complexity from the S3 approach scaled up with the production of between sequence length and the number of input channels in a polynomial manner. In comparison, NORMWEAR's final optimal channel-aware encoding schema (Luo et al., 2024), which is also leveraged in this study, have the complexity of

$$M_{\{NormWear, Pangu\_TS\}} = O(d \cdot C^2) \tag{10}$$

which only scale up with the input number of variates. Lastly, PANDA (Lai et al., 2025) proposed a very similar channel-aware mechanism, which the self-attention is applied on the input variate dimension across data representation from all time series. Such design has complexity equivalent to the *Cross-Attention* as analyzed by Luo et al. (2024):

$$M_{Panda} = O(d \cdot L \cdot C^2) \tag{11}$$

Finally, we can conclude that

$$M_{Sundial} > M_{Panda} > M_{\{NormWear, Pangu\_TS\}} \tag{12}$$

## G    EVALUATING DATA BALANCE

To evaluate the extent of data balance in terms of the metrics from chaotic theory as leveraged in this study, we mainly consider two main aspects, namely homogeneity and granularity of the distribution as presented in Figure 1 section B. Varied approaches for inspecting the balance attributes are presented below.

### G.1 UNNORMALIZED SHANNON ENTROPY.

Shannon entropy (Shannon, 1948) is one of the metrics widely used to evaluate the amount of information within a probability distribution:

$$H(p) = -\sum_{i=0}^{n} p_i \log(p_i) \tag{13}$$

where $p$ denote a set of probability sum up to 1. Such entropy value not only reflect homogeneity of a distribution, but also comprise granularity information, which is indicated by the fact that the more group of system that can be clustered from a dataset, the more likely the higher the value of $H(p)$.

### G.2 WEIGHTED SUM OF NORMALIZED SHANNON ENTROPY AND GRANULARITY.

Since normalized Shannon entropy (with denominator of $\log(n)$) is often used under different scenario, we need to have a separate metric to explicitly evaluate the extent of granularity of a distribution. To achieve this, we leveraged a straightforward scoring formula that reflect the relative granularity:

$$G(p) = \frac{|p|}{\max_{p' \in P} |p'|} \tag{14}$$

where $P$ is the collection of all the distributions in comparison, and $|p|$ indicate the number of bins or possible outcome of a particular distribution. The final balance score $B(p)$ is then expressed as a weighted sum of:

$$B(p) = \alpha \cdot \frac{H(p)}{\log(|p|)} + (1 - \alpha) \cdot G(p) \tag{15}$$

Since the value range of $H(p)$ and $G(p)$ is different, with relative ratio of total scores of around $4 : 6$, we use $\alpha = 0.6$ to balance this metric.

### G.3 WEIGHTED SUM OF COEFFICIENT OF VARIATION AND GRANULARITY.

Another commonly used metric that also reflect the extent of homogeneity of a distribution is coefficient of variation (Everitt & Skrondal, 2010), which is built on an assumption that the distribution is approximately a Gaussian distribution. From Figure 1, we observed that most of the distribution is nearly a Gaussian distribution with different mean and variance. We then leverage this new metric similar to the metric in section G.2, with $\frac{H(p)}{\log(|p|)}$ replaced by $\frac{1}{CV(p)}$, where $CV(p)$ is defined by:

$$CV(p) = \frac{\sigma_p}{\mu_p} \tag{16}$$

Similarly, we use $\alpha = 0.5$ for the same reason in section G.2 to balance the metric.

## H DETAILED EVALUATION PERFORMANCE

Table 6: **Main Evaluation Performance.** All the generative evaluation tasks are zero-shot forecasting on multivariate time series. The averaged results across test scenarios within each domain and each task are reported. For all the generative tasks, a lower MSE or MAE indicates a better prediction (Liu et al., 2025). For Battery SoH downstream tasks, mean absolute percentage error and $R^2$ are reported as the main metrics (Tan et al., 2025). For the wearable downstream tasks, AUC-ROC is reported as the main metric (Luo et al., 2024). $1^{st}$ Count represents the number of wins achieved by a model across all domains and test scenarios.

(a) **Generative Task Results**

| Evaluation Scenario | SUNDIAL$_{large}$ | | PANDA | | PANGU-TS (Ours) | | PANGU-TS Chaotic only | | PANGU-TS Sensor only | |
|---|---|---|---|---|---|---|---|---|---|---|
| **Metric** | MAE↓ | MSE↓ | MAE | MSE | MAE | MSE | MAE | MSE | MAE | MSE |
| Chaotic Short Forecast | 0.881 | 1.222 | **0.486** | **0.587** | 0.678 | 0.893 | 0.652 | 0.838 | 0.904 | 1.342 |
| Chaotic Long Forecast | 0.870 | 1.175 | **0.610** | **0.785** | 0.770 | 1.040 | 0.750 | 0.990 | 0.920 | 1.380 |
| Chaotic Short Simulation | 0.874 | 1.198 | **0.397** | **0.464** | 0.629 | 0.787 | 0.649 | 0.839 | 0.900 | 1.344 |
| Chaotic Long Simulation | 0.860 | 1.155 | **0.610** | **0.785** | 0.770 | 1.040 | 0.730 | 0.925 | 0.935 | 1.435 |
| Chaotic Generative Avg. | 0.889 | 1.241 | **0.510** | **0.638** | 0.645 | 0.833 | 0.631 | 0.796 | 0.896 | 1.319 |
| Wearable Short Forecast | 0.930 | 2.257 | 0.852 | 2.236 | **0.723** | **1.555** | 0.743 | 1.631 | 0.765 | 1.670 |
| Wearable Long Forecast | 0.950 | 2.170 | 1.010 | 2.750 | **0.880** | **1.950** | 0.880 | 1.960 | 0.950 | 2.330 |
| Wearable Short Simulation | 0.924 | 2.109 | 0.855 | 1.885 | **0.790** | **1.789** | 0.793 | 1.821 | 0.840 | 1.932 |
| Wearable Long Simulation | 0.970 | 2.260 | 0.950 | 2.180 | **0.900** | **2.070** | 0.900 | 2.120 | 0.960 | 2.300 |
| Wearable Generative Avg. | 0.953 | 2.255 | 0.840 | 1.933 | **0.740** | **1.538** | 0.810 | 1.813 | 0.835 | 1.883 |
| Civil Short Forecast | 0.539 | 0.576 | **0.413** | **0.405** | 0.503 | 0.561 | 0.492 | 0.535 | 0.533 | 0.590 |
| Civil Long Forecast | 0.654 | 0.953 | **0.557** | **0.778** | 0.636 | 0.938 | 0.633 | 0.925 | 0.678 | 1.039 |
| Civil Short Simulation | 0.569 | 0.614 | 0.509 | **0.516** | 0.518 | 0.569 | **0.508** | 0.549 | 0.582 | 0.670 |
| Civil Long Simulation | 0.675 | 0.961 | **0.619** | **0.816** | 0.640 | 0.925 | 0.632 | 0.897 | 0.697 | 1.061 |
| Civil Generative Avg. | 0.609 | 0.776 | **0.525** | **0.628** | 0.574 | 0.748 | 0.566 | 0.727 | 0.622 | 0.840 |
| Battery Short Forecast | 0.926 | 1.311 | 0.616 | 0.600 | **0.512** | **0.477** | 0.633 | 0.695 | 0.951 | 1.306 |
| Battery Long Forecast | 0.947 | 1.292 | 0.702 | 0.749 | **0.671** | 0.749 | 0.777 | 0.954 | 1.005 | 1.428 |
| Battery Short Simulation | 0.791 | 0.967 | 1.053 | 1.628 | **0.610** | **0.642** | 0.707 | 0.770 | 0.977 | 1.377 |
| Battery Long Simulation | 0.833 | 0.997 | 0.956 | 1.333 | **0.715** | **0.759** | 0.796 | 0.914 | 1.007 | 1.441 |
| Battery Generative Avg. | 0.874 | 1.141 | 0.832 | 1.077 | **0.627** | **0.657** | 0.728 | 0.833 | 0.985 | 1.388 |
| Short Forecast All. | 0.819 | 1.341 | **0.592** | 0.957 | 0.604 | **0.871** | 0.630 | 0.925 | 0.788 | 1.227 |
| Long Forecast All. | 0.855 | 1.397 | **0.720** | 1.265 | 0.740 | **1.174** | 0.762 | 1.216 | 0.875 | 1.470 |
| Short Simulate All. | 0.789 | 1.222 | 0.704 | 1.123 | **0.637** | **0.947** | 0.664 | 0.995 | 0.825 | 1.331 |
| Long Simulate All. | 0.834 | 1.343 | 0.784 | 1.278 | **0.738** | **1.149** | 0.761 | 1.192 | 0.884 | 1.499 |
| Generative Micro Avg. | 0.780 | 1.085 | 0.668 | 0.915 | **0.638** | **0.841** | 0.672 | 0.900 | 0.828 | 1.234 |

(b) **Downstream Task Results**

| Metric | MAPE↓ | $R^2$↑ | MAPE | $R^2$ | MAPE | $R^2$ | MAPE | $R^2$ | MAPE | $R^2$ |
|---|---|---|---|---|---|---|---|---|---|---|
| Battery Li-ion Downstream | 0.154 | 0.040 | 0.198 | -0.726 | **0.145** | **0.229** | 0.202 | -0.342 | 0.158 | 0.084 |
| Battery Na-ion Downstream | 0.050 | 0.954 | 0.075 | 0.879 | **0.047** | **0.963** | 0.070 | 0.888 | 0.051 | 0.947 |
| Battery Zn-coin Downstream | 1.550 | 0.551 | 1.321 | 0.255 | **0.685** | **0.563** | 0.964 | 0.447 | 0.695 | 0.371 |
| Battery Downstream Micro Avg. | 0.628 | 0.500 | 0.717 | 0.073 | 0.366 | **0.531** | 0.396 | 0.290 | **0.310** | 0.328 |
| **Metric** | AUC ROC↑ | | AUC ROC | | AUC ROC | | AUC ROC | | AUC ROC | |
| Wearable State Recognition | 0.790 | | 0.797 | | **0.813** | | 0.799 | | 0.790 | |
| Wearable EEG Tasks | **0.888** | | 0.856 | | 0.872 | | 0.862 | | 0.864 | |
| Wearable Vital Sign (1-MAPE) | 0.948 | | **0.954** | | 0.953 | | 0.949 | | 0.953 | |
| Wearable Disease Risk | 0.720 | | 0.709 | | **0.759** | | 0.741 | | 0.694 | |
| Wearable Downstream Avg. | 0.826 | | 0.814 | | **0.838** | | 0.826 | | 0.810 | |
| $1^{st}$ **Count** | 12 | | 23 | | **30** | | 5 | | 4 | |

Table 7: Relative Drop in MAE (%) and Average Drift across different methods between long term and short term generative series. From the results, no consistent pattern is observed across models. For instance, Sundial shows the least horizontal drift between short-term and long-term predictions, but its absolute generative error remains higher than other approaches. Moreover, the drift speeds for all methods are on the order of $10^{-4}$, indicating that error accumulation over these horizons is minimal.

| Metric | Sundial | Panda | Pangu-TS | Pangu-TS Chaotic only | Pangu-TS Sensor only |
|---|---|---|---|---|---|
| Chaotic Drop | +0.014% | -38.165% | -13.208% | -13.610% | -2.162% |
| Wearable Drop | -3.560% | -14.821% | -16.920% | -15.690% | -12.710% |
| Civil Drop | -19.955% | -27.611% | -24.994% | -26.393% | -23.223% |
| Battery Drop | -3.610% | +0.659% | -23.546% | -17.491% | -4.331% |
| **Avg Drop** | -6.439% | -19.985% | -19.667% | -18.296% | -10.607% |
| Chaotic Drift | $4.7 \times 10^{-5}$ | $6.6 \times 10^{-4}$ | $3.4 \times 10^{-4}$ | $3.5 \times 10^{-4}$ | $7.6 \times 10^{-5}$ |
| Wearable Drift | $1.3 \times 10^{-4}$ | $4.9 \times 10^{-4}$ | $5.0 \times 10^{-4}$ | $4.7 \times 10^{-4}$ | $4.0 \times 10^{-4}$ |
| Civil Drift | $4.3 \times 10^{-4}$ | $5.0 \times 10^{-4}$ | $5.0 \times 10^{-4}$ | $5.2 \times 10^{-4}$ | $5.1 \times 10^{-4}$ |
| Battery Drift | $1.2 \times 10^{-4}$ | $2.1 \times 10^{-5}$ | $5.2 \times 10^{-4}$ | $4.6 \times 10^{-4}$ | $1.6 \times 10^{-4}$ |
| **Avg Drift** | $1.6 \times 10^{-4}$ | $4.1 \times 10^{-4}$ | $4.6 \times 10^{-4}$ | $4.5 \times 10^{-4}$ | $2.9 \times 10^{-4}$ |

# I DETAILED DOWNSTREAM PERFORMANCE ON DIGITAL HEALTHCARE

Table 8: **Detailed performance on various downstream wearable-signal-based health related applications under full-shot linear probing evaluation.** The signal name, in column "Modality-Specific", following each performance score denotes the model specialized for that modality: PPG (Pillai et al., 2024), ECG-FM (McKeen et al., 2024), and EEG (Wang et al., 2025).

| Downstream Tasks | Pangu-TS | Panda | Sundial | Modality-Specific | NORMWEAR |
|---|---|---|---|---|---|
| WESAD | 72.524 | 73.187 | 70.529 | 56.656(max(PPG, ECG)) | **76.060** |
| UCI-HAR | 98.141 | 97.896 | 96.522 | - | **98.954** |
| DriverFatigue | 73.178 | 68.116 | 70.034 | **80.430(EEG)** | 74.292 |
| **Activity Recognition Avg.** | 81.281 | 79.733 | 79.028 | - | **83.102** |
| Epilepsy (eye open state) | **93.676** | 89.958 | 95.797 | 90.436(EEG) | 92.743 |
| Epilepsy (eye relaxation) | 96.639 | 94.085 | **97.390** | 95.552(EEG) | 94.828 |
| Epilepsy (health area) | 90.079 | 89.047 | **91.812** | 88.065(EEG) | 88.541 |
| Epilepsy (tumor area) | 88.257 | 86.415 | **91.103** | 87.258(EEG) | 87.197 |
| Epilepsy (seizure) | 99.339 | 98.636 | **99.723** | 94.616(EEG) | 97.053 |
| GAMEEMO | 54.946 | 55.263 | **56.814** | 55.420(EEG) | 54.937 |
| **EEG Main Tasks Avg.** | 87.156 | 85.567 | **88.773** | 85.225 | 85.883 |
| ECG-Abnormal | 98.605 | 99.542 | **99.73** | 89.898(ECG) | 99.140 |
| PPG-BP (HTN) | 62.268 | 56.082 | 55.47 | 61.839(PPG) | **62.341** |
| PPG-BP (DM) | **62.087** | 54.992 | 58.033 | 55.668(PPG) | 55.893 |
| PPG-BP (CVA) | 70.347 | 51.875 | 59.514 | **73.125(PPG)** | 70.625 |
| PPG-BP (CVD) | 62.021 | **63.121** | 59.275 | 49.066(PPG) | 51.773 |
| PhysioNet EMG | **99.999** | 99.948 | **99.999** | - | 99.216 |
| **Risk Evaluation Avg.** | **75.888** | 70.927 | 72.004 | - | 73.165 |
| Noninvasive-BP | **93.100** | 92.907 | 90.857 | 90.596(PPG) | 92.420 |
| PPG-Hgb | 93.779 | 94.451 | 94.419 | **94.912(PPG)** | 94.632 |
| Fetal-fPCG | 98.990 | 98.884 | **99.082** | - | 99.072 |
| **Vital Signs Avg.** | 95.290 | **95.414** | 94.786 | - | 95.375 |
| **Micro Avg.** | **83.776** | 81.356 | 82.561 | - | 82.762 |
| **Macro Avg.** | **84.904** | 82.910 | 83.648 | - | 84.381 |

## J ABLATION STUDY: SCALING IN SUBSET WITH VARIED BALANCE SCORES

Table 9: **Generative Results of Ablation Studies on Scaling in Pre-train Subsets with Varied Balance Scores.**

| Data Size | $10^3$ | | | | | | $10^4$ | | | | | | $10^5$ | | | | | |
|---|---|---|---|---|---|---|---|---|---|---|---|---|---|---|---|---|---|---|
| Evaluation Scenario | Balance | | Chaotic | | Sensor | | Balance | | Chaotic | | Sensor | | Balance | | Chaotic | | Sensor | |
| Metric | MAE | MSE | MAE | MSE | MAE | MSE | MAE | MSE | MAE | MSE | MAE | MSE | MAE | MSE | MAE | MSE | MAE | MSE |
| Chaotic Generative Avg. | 0.830 | 1.064 | 0.803 | 1.031 | 1.051 | 1.791 | 0.698 | 0.892 | 0.746 | 0.962 | 1.079 | 1.891 | 0.696 | 0.900 | 0.695 | 0.898 | 0.912 | 1.369 |
| Wearable Generative Avg. | 0.835 | 1.829 | 0.841 | 1.822 | 0.960 | 2.218 | 0.826 | 1.815 | 0.827 | 1.809 | 0.935 | 2.195 | 0.821 | 1.836 | 0.828 | 1.870 | 0.854 | 1.930 |
| Civil Generative Avg. | 0.608 | 0.786 | 0.595 | 0.762 | 0.731 | 1.125 | 0.565 | 0.717 | 0.575 | 0.738 | 0.715 | 1.105 | 0.574 | 0.748 | 0.566 | 0.727 | 0.622 | 0.840 |
| Battery Generative Avg. | 0.911 | 1.075 | 0.875 | 1.007 | 0.980 | 1.337 | 0.846 | 0.975 | 0.833 | 0.960 | 1.028 | 1.462 | 0.627 | 0.657 | 0.728 | 0.833 | 0.985 | 1.388 |
| Short Forecast All. | 0.752 | 1.067 | 0.740 | 1.050 | 0.910 | 1.557 | 0.674 | 0.933 | 0.690 | 0.969 | 0.898 | 1.505 | 0.604 | 0.871 | 0.630 | 0.925 | 0.788 | 1.227 |
| Long Forecast All. | 0.836 | 1.324 | 0.817 | 1.262 | 0.956 | 1.705 | 0.771 | 1.191 | 0.787 | 1.216 | 0.959 | 1.717 | 0.740 | 1.174 | 0.762 | 1.216 | 0.875 | 1.470 |
| Short Simulate All. | 0.770 | 1.105 | 0.751 | 1.069 | 0.912 | 1.536 | 0.705 | 1.018 | 0.710 | 1.016 | 0.924 | 1.596 | 0.637 | 0.947 | 0.664 | 0.995 | 0.825 | 1.331 |
| Long Simulate All. | 0.827 | 1.259 | 0.806 | 1.241 | 0.943 | 1.673 | 0.785 | 1.258 | 0.793 | 1.268 | 0.976 | 1.834 | 0.738 | 1.149 | 0.761 | 1.192 | 0.884 | 1.499 |
| Generative Micro Avg. | 0.779 | 1.036 | 0.757 | 0.996 | 0.900 | 1.422 | 0.715 | 0.943 | 0.723 | 0.957 | 0.915 | 1.477 | 0.638 | 0.841 | 0.672 | 0.900 | 0.828 | 1.234 |

Table 10: Observation of balance law in general (Univariate backbone). All values reported as MAE.

| | Pretrain Subset (Avg. balance) | | |
|---|---|---|---|
| **Balance Scores** | **0.891** | **1.003** | **1.139** |
| Short-term forecast | $0.768 \pm 0.030$ | $\underline{0.678 \pm 0.030}$ | $\mathbf{0.564 \pm 0.019}$ |
| Long-term forecast | $0.834 \pm 0.031$ | $\underline{0.764 \pm 0.031}$ | $\mathbf{0.694 \pm 0.022}$ |
| Short-term simulate | $0.767 \pm 0.027$ | $\underline{0.690 \pm 0.026}$ | $\mathbf{0.569 \pm 0.012}$ |
| Long-term simulate | $0.831 \pm 0.031$ | $\underline{0.772 \pm 0.026}$ | $\mathbf{0.706 \pm 0.019}$ |

Table 11: Observation of balance law accompanying scaling law using alternative backbones. All values reported as MAE.

| | Data Size | | |
|---|---|---|---|
| **Model / Balance** | $10^3$ | $10^4$ | $10^5$ |
| **Univariate backbone** | | | |
| Avg. balance = 1.003 | $\mathbf{0.749 \pm 0.029}$ | $0.743 \pm 0.032$ | $0.726 \pm 0.028$ |
| Avg. balance = 1.139 | $0.778 \pm 0.028$ | $\mathbf{0.741 \pm 0.034}$ | $\mathbf{0.633 \pm 0.021}$ |
| **Group attention backbone** | | | |
| Avg. balance = 1.003 | $\mathbf{0.673 \pm 0.046}$ | $\mathbf{0.629 \pm 0.049}$ | $0.564 \pm 0.045$ |
| Avg. balance = 1.139 | $0.796 \pm 0.029$ | $0.667 \pm 0.053$ | $\mathbf{0.547 \pm 0.039}$ |

## K   QUALITATIVE VISUALIZATION

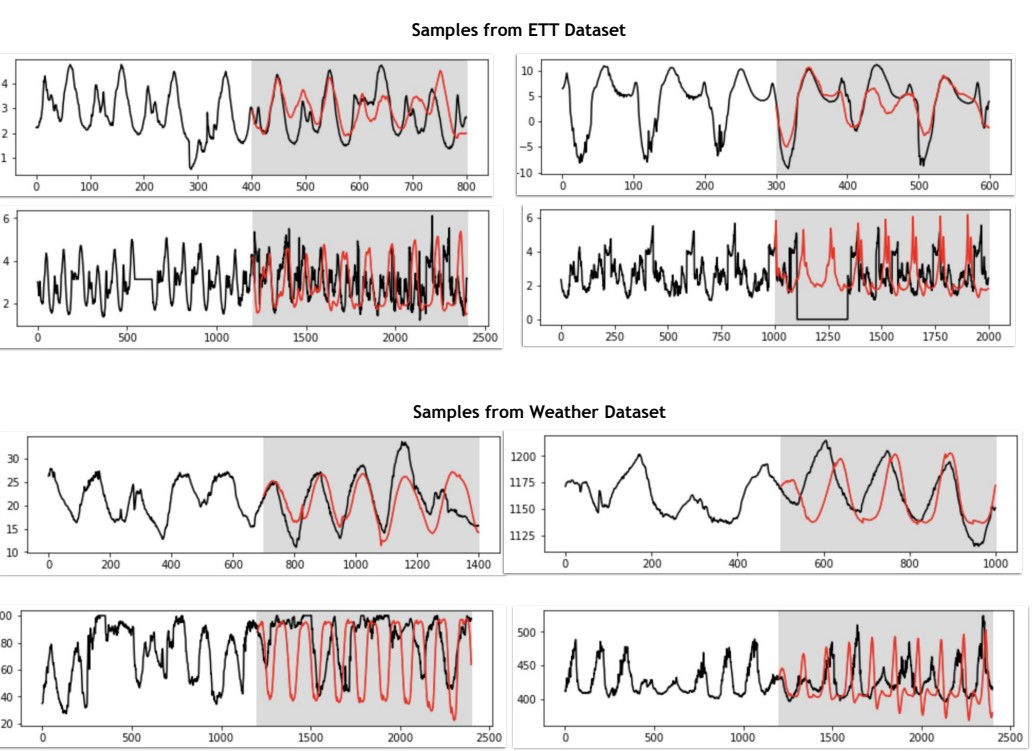

Figure 8: **Visualization of generation on time series randomly generated from civil monitoring datasets proposed by Wu et al. (2021).**

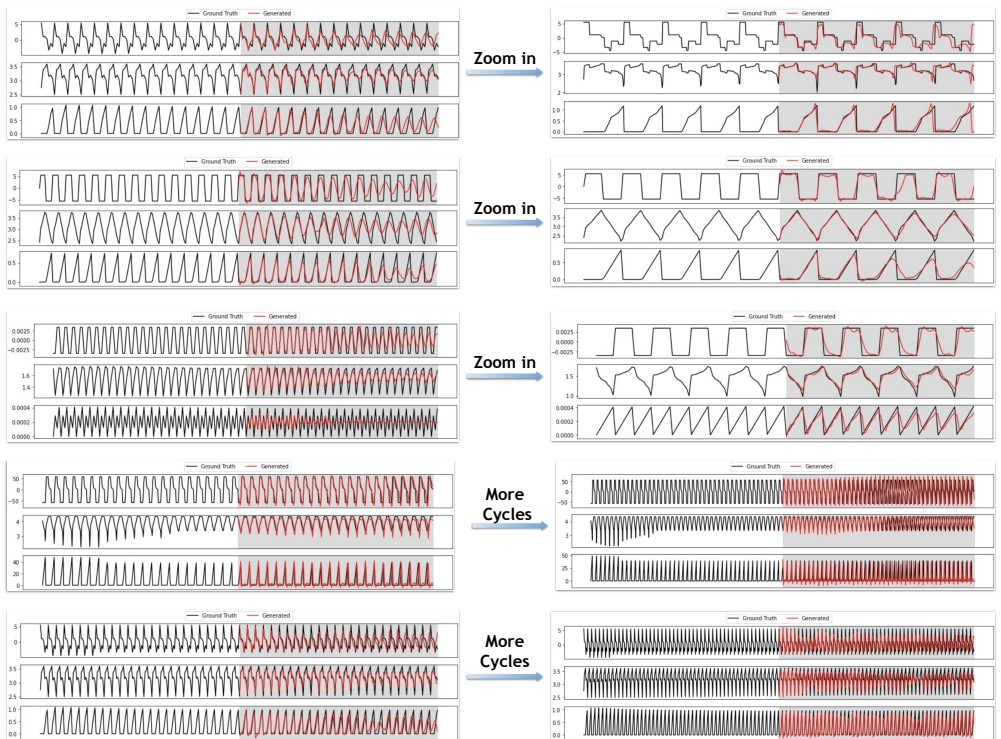

Figure 9: **Visualization of generation on time series randomly generated from battery datasets proposed by Tan et al. (2025).**

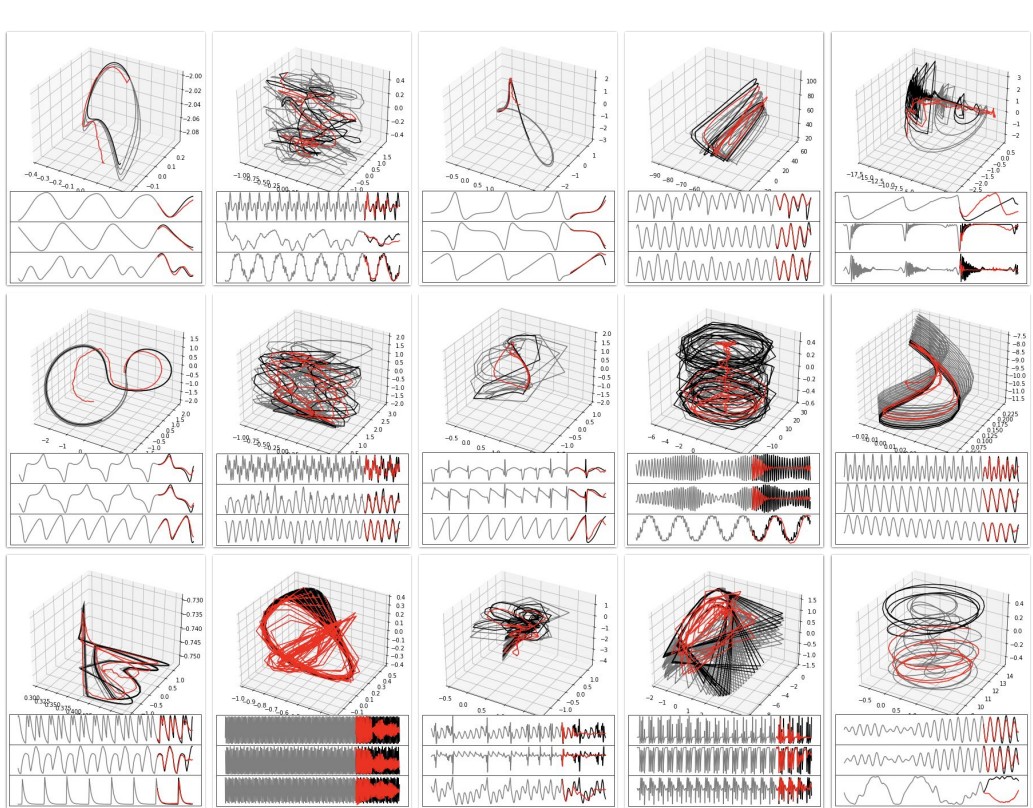

Figure 10: **Visualization of generation on time series randomly generated from chaotic system datasets proposed by Gilpin (2021).**

