# OpenReview forum: "Balanced Scaling Using Nonlinear Dynamic Metrics in Multivariate Time Series Modeling"
_ICLR.cc/2026/Conference — Submitted to ICLR 2026_

### Official Review · Reviewer_ArMM · 2025-10-29

**Soundness:** 2
**Presentation:** 2
**Contribution:** 3
**Rating:** 2
**Confidence:** 3

**Summary:**

The authors propose a method to estimate the “chaos” of a set of datasets containing time series data, using three metrics (DFA, LE and PE). Then, the authors present a foundation model trained on a dataset collection curated according to their proposed metrics. From their results, the authors show that their time series foundation model outperforms two existing foundation models, Sundial and Panda.

**Strengths:**

- The authors use three existing metrics to estimate 'chaos balance' inside pre-training datasets for time series foundation models. They show (Fig 5 and 6) how a higher “chaos” (according to their metrics) leads to improved performance in generative tasks, when using their proposed foundation model. This research direction is interesting.

- The proposed foundation model outperforms two existing baselines, Sundial and Pandas, on the presented tasks.

**Weaknesses:**

Major:

- **W1** - Experimental evaluation could be improved:
     - From the experiment, it was not clear to me whether their foundation model architecture is indeed outperforming existing approaches due to novel methodological implementations or due to the data.
     - All reported results lack an estimate of experimental uncertainty, i.e., variability across random initializations with different seeds. This is critical to ensure significance of the results and all associated claims.
     - Comparison is limited to two baselines.

- **W2** - To me, the interesting aspect of the paper was the impact of the data balance score on a foundation model's performance. However, this is limited to a single ablation study, appearing more as an addendum than the focus of the paper. Overall, the paper scope, as presented, looks limited and a bit “over the place”. It’s not clear if the paper is more about the analysis of the chaos metrics or about proposing a foundation model (of which the novelty could be argued).

- **W3** - The methodological description in Section 2.1 is often unclear. In particular, from the text, it wasn't clear to me where Equation 4 comes from. Also, the discrete Fourier series is not appropriate for general time series data, and the authors should be better off starting from the continuous Fourier transform.

- **W4** - The authors base their foundation model on an existing architecture, which is, however, not peer-reviewed and, indeed, not accepted for publication upon review by peers ([https://openreview.net/forum?id=4x83oH6Oy6](https://openreview.net/forum?id=4x83oH6Oy6 "https://openreview.net/forum?id=4x83oh6oy6")).


Minor:

- Fig. 3 is very small, with many unnecessary elements and icons. Fig.4 is also very small.

**Questions:**

- **Q1** (related to W1) - Is the superiority of PANGU-TS w.r.t. PANDA statistically significant?

- **Q2** (related to W1) - Would the authors consider re-training other existing foundation models, using the same strategy and datasets, in order to provide more generalizability to their results?

- **Q3** - Could the authors elaborate on the advantages of "chaos balance" apart from improved accuracy?

---

> ### Author Response · Authors · 2025-11-20
> **Response to Reviewer ArMM**
>
> **Weakness 1: Experimental evaluation could be improved:**
>
>
> We thank the reviewer for raising this point. Regarding the question on performance contribution, we apologize that we didn’t deliver the details of the ablation studies with controlled factors properly in the original version. We have included a detailed description in the general response “Corrected description of ablation studies”. We hope these additional details on the experimental setting could better support our claimed observation and model behavior.
>
> Regarding the performance variability, we did compute the variance of the generation performance across different task settings and domains, and we apologize for not presenting these results clearly in the original submission. We have updated the results in the main section of the revised manuscript as Table 1 and 2. We hope these reported statistics could provide better trust toward the robustness and replicability of the proposed framework.
>
> Regarding random seeds, since all generation tasks are performed in a zero-shot setting and we use the mean output of models as the final result. For downstream tasks, linear probing is solved using Newton’s conjugate gradient method for classification and Cholesky’s method for regression. As these approaches yield near closed-form solutions, we verified that changing the random seed does not alter the final outputs or performance and the variance across sub-task settings within each sub-domains.
>
> **Weakness 2: To me, the interesting aspect of the paper was the impact of the data balance score on a foundation model's performance. However, this is limited to a single ablation study, appearing more as an addendum than the focus of the paper. Overall, the paper scope, as presented, looks limited and a bit “over the place”. It’s not clear if the paper is more about the analysis of the chaos metrics or about proposing a foundation model (of which the novelty could be argued).**
>
> We thank the reviewer for indicating this important aspect. We apologize that we didn’t deliver the details of the core ablation studies with controlled factors (e.g. data scale, balance score, pretrain setting, etc.) properly in the original version. We have included a detailed description in the general response “Corrected description of ablation studies”, along with additional experiments in **general response “Ablation Study: Alternative Backbones”**. We hope these additional details on the experimental setting could better support our claimed observation and model behavior.
>
> **Weakness 3: The methodological description in Section 2.1 is often unclear. In particular, from the text, it wasn't clear to me where Equation 4 comes from. Also, the discrete Fourier series is not appropriate for general time series data, and the authors should be better off starting from the continuous Fourier transform.**
>
> We thank the reviewer for raising this point. The Equation (4) as discussed with Platonic representation hypothesis (PRH) stands more as a motivating assumption rather than a direct guidance toward the core methodology. In the revised manuscript, we have clarified the intended role of PRH and with what form it appears in time-series space in terms of describing varied data augmentation method in current literature, and motivation toward exploring balance law, as summarized in the general response titled “Concerns regarding Platonic Representation Hypothesis and balance law motivation.” We hope these revisions could introduce better clarity around the roles of these notions involved in our study.
>
> In addition, we would like to clarify that the Fourier series approximation discussed in our work is conceptually different from the Fourier transform. In this context, the distinction between discrete and continuous does not directly apply. We hope this clarification helps to better convey the motivation behind our discussion.
>
> **Weakness 4: The authors base their foundation model on an existing architecture, which is, however, not peer-reviewed and, indeed, not accepted for publication upon review by peers (https://openreview.net/forum?id=4x83oH6Oy6).**
>
> We would like to clarify that our architecture design is not solely based on the one mentioned by the reviewer. While we respect the reviewer’s perspective, we note that the value of prior work should not be determined solely by its publication status. The work referenced by the reviewer has received growing recognition within the community, and more importantly, their study provides thorough complexity analysis, solid empirical evidence, and a publicly available codebase, also as widely acknowledged by the corresponding reviewers in the forum shared by the reviewer.

---

> > ### Author Response · Authors · 2025-11-20
> > **Response to Reviewer ArMM**
> >
> > **Minor Weakness :Fig. 3 is very small, with many unnecessary elements and icons. Fig.4 is also very small.**
> >
> > We thank the reviewer for pointing the issue out. We have adjusted the font size of Figure 3 and 4, and we hope that could help with the visual effect of the figure. We’d also like to respectfully disagree with the reviewer that Figure 3 contains unnecessary elements. Downstream inference is one of the core aspects to evaluate the quality of the learned representation, and the tasks accompanied with icons is an exact one-on-one match to the diverse downstream tasks involved in our method evaluation.
> >
> > **Question 1: (related to W1) - Is the superiority of PANGU-TS w.r.t. PANDA statistically significant?**
> >
> >
> > We appreciate the reviewer’s careful attention to this point. Yes, the performance difference is statistically significant. The corresponding test results are reported in Figure 3, panel B, which we now highlight more clearly in the revised manuscript. We thank the reviewer for prompting us to better clarify this aspect.
> >
> > **Question 2 (related to W1) - Would the authors consider re-training other existing foundation models, using the same strategy and datasets, in order to provide more generalizability to their results?**
> >
> > We thank the reviewer for this thoughtful suggestion. We fully agree with the importance of assessing generalizability across architectures. To this end, we have reproduced the core experiments using alternative backbone models. The corresponding results are now provided in the **general response “Ablation Study: Alternative Backbones.”** We appreciate the reviewer’s insight, which helped us clarify and better highlight these additional experiments.
> >
> > **Question 3 Could the authors elaborate on the advantages of "chaos balance" apart from improved accuracy?**
> >
> > We thank the reviewer for this thoughtful comment. We have added a concise statement in the discussion section highlighting the advantages of chaos balance beyond improved accuracy. The updated text is provided below:
> >
> > Apart from improved accuracy as extensively justified through this study, chaos based balance inspection essentially provides a principled and interpretable way to assess and control the diversity of dynamical systems in pretraining datasets. It offers insights into dataset quality, and complements scaling by ensuring the model captures representative patterns rather than being dominated by over-represented dynamics.

---

> > > ### Author Response · Authors · 2025-11-20
> > > **Response to Reviewer ArMM**
> > >
> > > We sincerely thank you for your insightful review and valuable suggestions, which have greatly contributed to improving our paper. We hope our responses fully address your concerns, and we would be delighted to continue the discussion if you have any further questions or comments.

---

### Official Review · Reviewer_qH8D · 2025-10-30

**Soundness:** 2
**Presentation:** 2
**Contribution:** 2
**Rating:** 4
**Confidence:** 2

**Summary:**

The paper introduces PANGU-TS, a multivariate time-series foundation model pretrained on a chaos-theoretic balanced dataset, inspired by the Platonic Representation Hypothesis. The authors argue that dataset balance—quantified via nonlinear dynamic metrics such as DFA, Lyapunov exponent, and Permutation entropy—is more critical than raw dataset scale for achieving generalizable temporal representations.
Extensive experiments across chaotic, wearable, battery, and infrastructure domains demonstrate strong zero-shot forecasting and representation transferability. The study’s novelty lies in applying chaos theory to dataset characterization and proposing that balanced dynamic diversity enhances pretraining quality.

**Strengths:**

- The paper introduces a fresh perspective by integrating chaos theory metrics into dataset evaluation, linking nonlinear dynamics with representation learning.
- PANGU-TS exhibits competitive or superior zero-shot forecasting compared to existing foundation models (e.g., Chronos, SUNDIAL, PANDA) across multiple real-world domains.
- The work provides early empirical evidence for the Platonic Representation Hypothesis, highlighting that qualitative balance may outweigh dataset size for robust pretraining.
- The authors detail their pretraining hyperparameters, masking strategies, and release plans for code and scripts, enhancing research transparency.
- The proposed framework bridges chaotic system modeling and AI foundation modeling, offering potential for wide application in scientific, industrial, and healthcare time-series settings.

**Weaknesses:**

- The *Platonic Representation Hypothesis* is explained vaguely. The paper does not clearly articulate how this philosophical concept operationally connects to model design or chaos metrics.
- The procedures for computing and combining chaos-based balance metrics (DFA, LE, PE) are insufficiently specified, making reproduction difficult.
- The claim that “balanced data matters more than sheer scale” lacks strong empirical justification. Figure 6 and related analyses require clearer numerical interpretation and stronger comparisons with large-scale baselines.
- Implementation specifics, such as the definition of the aggregation operator (∘), [MASK] token training dynamics, and Conv1D multivariate treatment, are underexplained.
- The paper overlooks potential biases and privacy risks when applying PANGU-TS to sensitive domains (such as healthcare) and provides minimal reflection on limitations or misuse.

**Questions:**

- How does the hypothesis translate into concrete architectural or training choices?
- What empirical or mathematical form does a “Platonic” representation take in time-series space?
- How was the weighting factor ($\alpha$) selected, and how sensitive are results to its value?
- Can you provide statistical evidence (e.g., correlation or p-values) supporting that balance, not scale, drives performance?
- What happens when model size or dataset volume is controlled while varying bthe alance?
- Could an ablation explicitly show performance variation with different balance vs. scale settings?
- How does Conv1D process multivariate inputs—independently per channel or through cross-channel fusion?

---

> ### Author Response · Authors · 2025-11-20
> **Response to Reviewer qH8D**
>
> **Weakness on insufficient balance metrics description: The procedures for computing and combining chaos-based balance metrics (DFA, LE, PE) are insufficiently specified, making reproduction difficult.**
>
> We thank the reviewer for highlighting the absence of a detailed description of the procedure for computing chaos-based balance metrics. In the revised version, we have included comprehensive pseudo-code in Appendix A, which outlines the full process from metric computation to the categorization of dynamical system types. The codebase is currently being cleaned and refined to ensure that the final version is as user-friendly as possible for future researchers and developers.
>
> **Weakness on insufficient method description: Implementation specifics, such as the definition of the aggregation operator (∘), [MASK] token training dynamics, and Conv1D multivariate treatment, are underexplained.**
>
> We appreciate the reviewer’s thoughtful comment. To address this, we have enriched Appendix D (Model Implementation Details) with explicit operational expressions in the revised version. We hope the expanded explanation brings greater clarity to the engineering side of the method.
>
> **Weakness on privacy concern and lack of limitation discussion: The paper overlooks potential biases and privacy risks when applying PANGU-TS to sensitive domains (such as healthcare) and provides minimal reflection on limitations or misuse.**
>
> We agree with the reviewer that applying PANGU-TS in sensitive domains, such as healthcare, may entail potential biases and privacy risks. Addressing these concerns is beyond the scope of the current study and warrants careful consideration in future work. We’d also like to apologize for including the wrong limitation paragraph in the original submission. We have corrected it in the revised version, and we hope that it could provide more contextual information regarding our reflection on limitations.
>
>
> **Question on Platonic Representation Hypothesis:**
> **- How does the hypothesis translate into concrete architectural or training choices?**
> **- What empirical or mathematical form does a “Platonic” representation take in time-series space?**
> **- How was the weighting factor () selected, and how sensitive are results to its value?**
>
> We thank the reviewer for raising this point. The Platonic representation hypothesis (PRH) stands more as a motivating assumption rather than a direct guidance toward architectural design. In the revised manuscript, we have clarified the intended role of PRH and with what form it appears in time-series space in terms of describing varied data augmentation method in current literature, and motivation toward exploring balance law, as summarized in the **general response titled “Concerns regarding Platonic Representation Hypothesis and balance law motivation.”** We hope these revisions could introduce better clarity around the roles of these notions involved in our study.
>
> **Question on Detailed Observation of Influence from Balancing dataset:**
> **- Can you provide statistical evidence (e.g., correlation or p-values) supporting that balance, not scale, drives performance?**
> **- What happens when model size or dataset volume is controlled while varying the balance?**
> **- Could an ablation explicitly show performance variation with different balance vs. scale settings?**
>
> We thank the reviewer for indicating this important aspect. We apologize that we didn’t deliver the details of the core ablation studies with controlled factors (e.g. data scale, balance score, pretrain setting, etc.) properly in the original version. We have included a detailed description in the general response “Corrected description of ablation studies”, along with additional experiments in **general response “Ablation Study: Alternative Backbones”**. We hope these additional details on the experimental setting could better support our claimed observation and model behavior.
>
> **Question on Conv1D: How does Conv1D process multivariate inputs—independently per channel or through cross-channel fusion?**
>
> Conv1D is applied independently on each channel, ensuring that the model is not sensitive to position.

---

> > ### Author Response · Authors · 2025-11-20
> > **Response to Reviewer qH8D**
> >
> > Thank you very much for your careful review and constructive suggestions. Your comments have been extremely helpful in refining our work. We hope the revisions adequately address your recommendations, and we remain open to any further discussion or feedback.

---

> ### Comment · Reviewer_qH8D · 2025-11-20
>
> Thank you for the opportunity to review the authors’ revised manuscript and detailed responses.
>
> After carefully examining the updated version, I believe that the authors have sufficiently addressed the key concerns raised in the original review:
>
> - Methodological ambiguities, such as the definition of the aggregation operator, [MASK] token training dynamics, and the handling of multivariate inputs in Conv1D, have been clarified in the expanded Appendix D.
> - The revised manuscript now includes a more appropriate and meaningful discussion of limitations and privacy considerations, correcting the earlier oversight regarding sensitive-domain deployment.
> - The role of the Platonic Representation Hypothesis has been clarified, and the revisions help position it correctly as a conceptual motivation rather than a prescriptive architectural constraint.
> - The newly added ablation descriptions and supplemental experiments address earlier concerns about distinguishing the influence of balance versus scale, providing more concrete evidence for the authors’ claims.
>
> Overall, the revisions improve the clarity, methodological transparency, and contextual framing of the work.
> I have raised the overall score.

---

> > ### Author Response · Authors · 2025-11-20
> > **Response to Reviewer qH8D**
> >
> > Dear Reviewer qH8D,
> >
> > Thank you very much for your encouraging feedback and for your positive reassessment of our work. We truly appreciate your thoughtful evaluation!
> >
> > Best regards,
> > Authors

---

### Official Review · Reviewer_nJHj · 2025-10-31

**Soundness:** 2
**Presentation:** 3
**Contribution:** 2
**Rating:** 4
**Confidence:** 3

**Summary:**

This paper introduces the Platonic representation hypothesis for the domain of multi-variance time series. Building on this hypothesis, the authors propose clustering data using metrics from chaotic theory to construct a more balanced multivariate time-series pre-training dataset. Based on this dataset, this paper introduces the PANGU-TS model, which achieves strong performance in zero-shot forecasting across diverse multivariate time-series scenarios, as well as on other downstream tasks.

**Strengths:**

To the best of my knowledge, this is the first work that attempts to establish a unified mathematical representation for multivariate time-series data in the context of time-series foundation models, and to propose a systematic data-mixing strategy based on it.
The experimental results demonstrate that, in time-series settings, a principled data-mixing scheme can be more critical for improving model performance than simply scaling up the data volume.

**Weaknesses:**

1. The transition from the Platonic representation hypothesis to using chaotic metrics for data balance assessment is entirely heuristic. The paper does not clarify the relationship between these chaotic metrics and the coefficients in Equation (4). It remains unclear how the Platonic representation hypothesis concretely supports the subsequent methodological design.
2. The core methodology and assumptions of the paper, such as the generality of Equation (4), the selection of chaotic metrics, and the specific clustering and mixing strategies, lack theoretical justification, and the overall approach relies heavily on empirical validation.
3. However, the experimental evidence is not sufficiently solid, with several issues:
  a. Baseline selection: The paper does not compare against some newer and stronger univariate foundation models. The justification provided is unconvincing, especially since the referenced works only compare against earlier or smaller univariate models. In addition, the authors do not evaluate against NormWear, which shares a similar architecture with PANGU-TS.
  b. Comparison with PANDA: On the chaotic evaluation dataset, PANGU-TS does not outperform PANDA, suggesting that the proposed data-mixing strategy may sacrifice performance in domain-specific settings.
  c. Ablation studies: The ablations are simplistic and only demonstrate that mixing data from different sources yields improvements; they do not establish the effectiveness of the specific mixing scheme proposed in the paper.

**Questions:**

1. In Figure 3.A, the connection arrows among the Encoder, Lightweight Decoder, and Transformer appear somewhat unclear. What is the correct input–output flow among these components?

---

> ### Author Response · Authors · 2025-11-20
> **Response to Reviewer nJHj**
>
> **Weakness 1:The transition from the Platonic representation hypothesis to using chaotic metrics for data balance assessment is entirely heuristic. The paper does not clarify the relationship between these chaotic metrics and the coefficients in Equation (4). It remains unclear how the Platonic representation hypothesis concretely supports the subsequent methodological design.**
>
> We thank the reviewer for raising this point. The Platonic representation hypothesis (PRH) stands more as a motivating assumption rather than a formal theorem. In the revised manuscript, we have clarified the intended role of PRH and its relationship to the Equation (4) , and motivation toward exploring balance law, as summarized in the **general response titled “Concerns regarding Platonic Representation Hypothesis and balance law motivation.”** We hope these revisions make the motivation of our study more explicit.
>
> **Weakness 2: The core methodology and assumptions of the paper, such as the generality of Equation (4), the selection of chaotic metrics, and the specific clustering and mixing strategies, lack theoretical justification, and the overall approach relies heavily on empirical validation.**
>
> We thank the reviewer for highlighting this point. As this study primarily aims to empirically investigate the extent to which the balancing law manifests in multivariate time series modeling, it is designed as an empirical study rather than a theoretically focused work. To provide greater clarity on the exact procedure for balance inspection, we have included detailed pseudo-code in **Appendix A** of the revised version, covering the full workflow from chaos-metric computation to final dynamical system categorization. Several references are also included in **Section 3.1** of the Methods, demonstrating that we follow established guidance from the chaos theory literature throughout the procedure for inspecting the balance score. We hope that these additions help to more clearly convey the core contributions of our work.
>
> **Weakness 3a: However, the experimental evidence is not sufficiently solid, with several issues: a. Baseline selection: The paper does not compare against some newer and stronger univariate foundation models.
> The justification provided is unconvincing, especially since the referenced works only compare against earlier or smaller univariate models.
> In addition, the authors do not evaluate against NormWear, which shares a similar architecture with PANGU-TS.**
>
> We thank the reviewer for raising the concern on the baseline comparison which is one of the very essential factors of the quality of the justification of our claims.
> - We would like to respectfully clarify that our comparison is not limited to earlier or univariate models. The baselines included in our experiments, Sundial, Panda, and NormWear, all support multivariate input–output modeling and represent state-of-the-art time series approaches among recent open-source works (Sundial released in 2025, Panda and NormWear first released in 2024).
> - Secondly, we also included NormWear in the baseline comparison for the downstream tasks. Since NormWear was not originally designed for generative modeling, we considered it as a strong baseline specifically for evaluating representation quality. The corresponding results were presented in **Appendix I**.
> - Finally, as stated in **Section 3.3 (Downstream Evaluation)**, we had explained our rationale for baseline selection, that prior studies have already established the superiority of multivariate approaches over univariate ones, which is not the primary focus of this work. Nevertheless, we strongly agree with the reviewer that including a univariate modeling approach could further clarify the generality of the proposed observations. We have reproduced the experiments with a uni-variate backbone, and we agree that this is a great suggestion to strengthen the generality of the observation, and to better contextualize the performance of the presented paradigm model. We have presented these additional results in the **general response “Ablation Study: Alternative Backbones”**.

---

> > ### Author Response · Authors · 2025-11-20
> > **Response to Reviewer nJHj**
> >
> > **Weakness 3b: Comparison with PANDA: On the chaotic evaluation dataset, PANGU-TS does not outperform PANDA, suggesting that the proposed data-mixing strategy may sacrifice performance in domain-specific settings.**
> >
> > We agree with the reviewer that Pangu-TS trained on the proposed balanced pretraining datasets may show reduced performance in certain sub-domains such as chaotic dynamics. However, we do not view this as a weakness but rather as an expected outcome of the balancing process, which inherently trades off specialization in domain-specific dynamics for improved generality across diverse dynamical regimes.
> >
> > The purpose of inspecting dataset balance, quantified through the balance presence of different types of dynamical systems, is precisely to understand how far such generality can extend. For instance, PANDA demonstrated that models trained solely on ODE-based data could generalize to PDE-like patterns, while NormWear showed that models trained on mixed-channel physiological signals could generalize to unseen sensor configurations. Similar behaviors are observed here: although Pangu-TS trained on balanced datasets performs slightly worse than PANDA on the chaotic domain, it achieves stronger generalization in wearable-related settings and demonstrates overall better generative and representational quality in aggregate evaluations and downstream tasks.
> >
> > We would also like to emphasize that our goal is not to rank models but to use such comparisons to contextualize the performance of the proposed paradigm. More importantly, this observation reinforces one of the central themes of our work: the presence of a **balancing law** that governs the tradeoff between domain specialization and cross-domain generality. We therefore see this not as a limitation, but as a valuable subtopic worth further discussion within the broader scope of the study, and we appreciate the reviewer for highlighting this important perspective. We have incorporated this into the limitation discussion in the revised version of the paper.
> >
> > **Weakness 3c: Ablation studies: The ablations are simplistic and only demonstrate that mixing data from different sources yields improvements; they do not establish the effectiveness of the specific mixing scheme proposed in the paper.
> >
> > We thank the reviewer for highlighting this important point. To clarify, our ablation studies are not simple mixtures of data from different sources; they are conducted under controlled conditions, including data size, balance score, and model backbones. We apologize for not presenting the details of these controlled ablation settings clearly in the original submission. A more thorough explanation has now been added in the **general response “Corrected description of ablation studies.”** We hope these additional details on the experimental setup can better support our stated observations and the model’s behavior.
> >
> > **Question 1: In Figure 3.A, the connection arrows among the Encoder, Lightweight Decoder, and Transformer appear somewhat unclear. What is the correct input–output flow among these components?**
> >
> >
> > We thank the reviewer for this question for clarification. The input-output flow is mainly represented by the thick arrow filled with light blue colors. The line-arrow serves more as a demonstration of details of a module. We have adjusted the font size and added a remark panel on the side of model architecture for better clarification. In Figure 3, we are trying to convey that though the attentions are applied on different data dimensions as mentioned in method section 3.2, the implementation of attention mechanism stays the same for intra- and inter-channel encoder and the lightweight decoder.

---

> > > ### Author Response · Authors · 2025-11-20
> > > **Response to Reviewer nJHj**
> > >
> > > We are truly grateful for your thoughtful and detailed review. Your constructive feedback has significantly enhanced our manuscript. We hope our responses meet your expectations, and we would be glad to discuss any additional points or suggestions you might have.

---

### Official Review · Reviewer_ho8Y · 2025-11-01

**Soundness:** 2
**Presentation:** 2
**Contribution:** 2
**Rating:** 2
**Confidence:** 3

**Summary:**

The paper argues that many time-series systems can be viewed as a composition of base signals plus timestamp-wise perturbations, and that balancing the distribution of dynamical types before pretraining improves generalization. Building on this, the authors propose PANGU-TS, a pretraining pipeline that quantifies dataset balance with nonlinear dynamical metrics (DFA, Lyapunov exponent, persistent entropy) and then trains a channel-aware MAE on a more balanced corpus.

**Strengths:**

1. Chaos-theory–based balance assessment: The use of DFA, Lyapunov exponent, and persistent entropy gives a clear diagnostic of “what kinds of dynamics” are present and how evenly they are covered, K-means with an elbow rule is used to visualize and quantify balance.

2. Domain diversity: Pretraining/evaluation spans heterogeneous sources (e.g., wearable/health, chaotic systems, battery, civil monitoring), aligning with the paper’s goal of modality-agnostic time-series foundations.

**Weaknesses:**

1. Limited theoretical grounding for PRH. The Platonic Representation Hypothesis is presented as a motivating assumption rather than a theorem; formal identifiability/equivalence guarantees are not established.

2. Training relies on MSE reconstruction only, there is no explicit spectral/topological or physics-consistency regularization, which could better preserve frequency-domain or structural properties.

3. Forecasting/simulation is framed generatively via masking, but the paper provides limited analysis of error accumulation, mode collapse, or noise amplification under long rollouts.

4. While many hyperparameters are listed, broader reporting (multiple seeds/variances, error bars) could strengthen claims of robustness and replicability.

**Questions:**

1. If we mix architecturally different backbones (CNN-style, non-autoregressive transformers, state-space models), do the learned representations still converge to a common latent space? What are potential counterexamples or boundary conditions for PRH in time series?

2. Have you tried scheduled sampling/teacher forcing or a denoising step during generation to mitigate error drift on long horizons? Any quantitative study on horizon length vs. degradation?

3. With total tokens/length/variates held constant, how much of the gain comes purely from balancing the dynamical mix vs. from adding more data? A strict controlled ablation would clarify causal impact.

---

> ### Author Response · Authors · 2025-11-20
> **Response to Reviewer ho8Y**
>
> **Weakness 1: Limited theoretical grounding for PRH. The Platonic Representation Hypothesis is presented as a motivating assumption rather than a theorem; formal identifiability/equivalence guarantees are not established.**
>
>
> We thank the reviewer for raising this point. We agree that PRH functions as a motivating assumption rather than a formal theorem. In the revised manuscript, we have clarified the intended role of PRH and its relationship to the balance law, as summarized in the **general response titled “Concerns regarding Platonic Representation Hypothesis and balance law motivation.”** We hope these revisions make the motivation of our study more explicit.
>
> **Weakness 2: Training relies on MSE reconstruction only, there is no explicit spectral/topological or physics-consistency regularization, which could better preserve frequency-domain or structural properties.**
>
> We agree with the reviewer that physics-informed modeling is an important and valuable research direction. However, it is not the primary focus of this study, which instead aims to investigate the impact of pretrained data quality and examine whether a balancing law accompanies the scaling law.
>
> **Weakness 3: Forecasting/simulation is framed generatively via masking, but the paper provides limited analysis of error accumulation, mode collapse, or noise amplification under long rollouts.**
>
> We agree with the reviewer that having quantitative analysis is important for inspecting model behaviors. Additional results are added in our response to question 2.
>
> **Weakness 4: While many hyperparameters are listed, broader reporting (multiple seeds/variances, error bars) could strengthen claims of robustness and replicability.**
>
> We thank the reviewer for raising this point. We did compute the variance of the generation performance across different task settings and domains, and we apologize for not presenting these results clearly in the original submission. We have updated the results in the main section of the **revised manuscript as Table 1 and 2**. We hope these reported statistics could provide better trust toward the robustness and replicability of the proposed framework.
>
> Regarding random seeds, since all generation tasks are performed in a zero-shot setting and we use the mean output of models as the final result. For downstream tasks, linear probing is solved using Newton’s conjugate gradient method for classification and Cholesky’s method for regression. As these approaches yield near closed-form solutions, we verified that changing the random seed does not alter the final outputs or performance and the variance across sub-task settings within each sub-domains.
>
> The entire codebase, including pretraining, evaluation, and analysis, will be made publicly available to ensure full reproducibility. The codebase is currently being cleaned and refined to ensure that the final version is as user-friendly as possible for future researchers and developers.

---

> > ### Author Response · Authors · 2025-11-20
> > **Response to Reviewer ho8Y**
> >
> > **Question 1: If we mix architecturally different backbones (CNN-style, non-autoregressive transformers, state-space models), do the learned representations still converge to a common latent space? What are potential counterexamples or boundary conditions for PRH in time series?**
> >
> > We thank the reviewer raising this inspiring question. To strengthen the observation of balancing law, which is the primary focus of this study, we provide additional results in **general response “Ablation Study: Alternative Backbones”**, which demonstrate that the observed balancing law is backbone-agnostic, holding true for both uni-variate backbones and those incorporating channel-wise attention. The pretraining setting remains controlled in terms of consistency in hyperparameters, data size, and training steps.
> >
> > Regarding the potential counterexamples, we agree that there are some potential boundaries for PRH in time series, for example, dynamical systems from different domains might be dominated by different sets of physical patterns. Rather than viewing it as a counterexample, we consider it a potential challenge that warrants further investigation, we have added a brief discussion in the **limitation section**, and we are sorry that we included a wrong paragraph to this section in the original submission.
> >
> > **Question 2: Have you tried scheduled sampling/teacher forcing or a denoising step during generation to mitigate error drift on long horizons? Any quantitative study on horizon length vs. degradation?**
> >
> > Thank you for bringing up this point. In this work, we did not apply scheduled sampling during generation. The model includes a deconvolution step at the end of the decoder, where the intermediate outputs can be viewed as different possible variations, which are then aggregated into a mean output. However, this design mainly serves as an architectural choice rather than a focus on generation dynamics study.
> >
> > Regarding the quantitative study on the horizon degradation, we agree with the reviewer that it is one of the representative aspects for inspecting model behavior, and we thank the reviewer for the opportunity to add additional analysis results. We present the result below:
> > - The percentage error drop is calculated as (short_term_generate_MAE - long_term_generate_MAE) / short_term_generate_MAE
> > - The drift speed is calculated as |short_term_generate_MAE - long_term_generate_MAE) | / time_step_difference
> >
> > | Metric  | Sundial    | Panda      | Pangu-TS   | Pangu-TS Chaotic Only | Pangu-TS Sensor Only |
> > |-------------------|------------|------------|------------|------------------------|------------------------|
> > | Chaotic Drop      | +0.014%    | -38.165%   | -13.208%   | -13.610% | -2.162%  |
> > | Wearable Drop     | -3.560%    | -14.821%   | -16.920%   | -15.690%  | -12.710% |
> > | Civil Drop        | -19.955%   | -27.611%   | -24.994%   | -26.393%  | -23.223%|
> > | Battery Drop      | -3.610%    | +0.659%    | -23.546%   | -17.491%  | -4.331%  |
> > | Avg Drop          | -6.439%    | -19.985%   | -19.667%   | -18.296%  | -10.607% |
> > | Chaotic Drift     | 4.7e-5     | 6.6e-4     | 3.4e-4     | 3.5e-4 | 7.6e-5    |
> > | Wearable Drift    | 1.3e-4     | 4.9e-4     | 5.0e-4     | 4.7e-4  | 4.0e-4   |
> > | Civil Drift       | 4.3e-4     | 5.0e-4     | 5.0e-4     | 5.2e-4   | 5.1e-4     |
> > | Battery Drift     | 1.2e-4     | 2.1e-5     | 5.2e-4     | 4.6e-4  | 1.6e-4    |
> > | Avg Drift         | 1.6e-4     | 4.1e-4     | 4.6e-4     | 4.5e-4    | 2.9e-4    |
> >
> > From the results, no consistent pattern is observed across models. For instance, Sundial shows the least horizontal drift between short-term and long-term predictions, but its absolute generative error remains higher than other approaches. Moreover, the drift speeds for all methods are on the order of 10^{-4}, indicating that error accumulation over these horizons is minimal. We agree that examining long-term drift is an important research direction, but it is not the primary focus of this study. These results are included in the **Appendix H** for reference.
> >
> > **Question 3: With total tokens/length/variates held constant, how much of the gain comes purely from balancing the dynamical mix vs. from adding more data? A strict controlled ablation would clarify causal impact.**
> >
> > We thank the reviewer for indicating this important aspect. We apologize that we didn’t deliver the details of the core ablation studies with controlled factors properly in the original version. We have included a detailed description in the **general response “Corrected description of ablation studies”**. We hope these additional details on the experimental setting could better support our claimed observation and model behavior.

---

> > > ### Author Response · Authors · 2025-11-20
> > > **Response to Reviewer ho8Y**
> > >
> > > We sincerely appreciate the insightful comments and constructive suggestions you provided, which have greatly helped us improve the quality of our paper. We hope our revisions address your concerns, and we warmly welcome any further questions or feedback you may have. Thank you again for your valuable input.

---

### Author Response · Authors · 2025-11-20
**Response to common concerns**

# Response to common concerns raised by reviewers:

We thank the reviewers for all the detailed comments and constructive feedback. We have noticed that there are 3 concerns that are discussed by all the reviewers, which are directly related to the core contribution of our study, and we thank the opportunity to clarify, improve writings, and present additional experiments. We present the responses here in 3 sections:

- Section 1: Concerns regarding Platonic Representation Hypothesis and balance law motivation
- Section 2: Corrected description of ablation studies
- Section 3: Ablation Study: Alternative Backbones

---

> ### Author Response · Authors · 2025-11-20
> **Section 1: Concerns regarding Platonic Representation Hypothesis and balance law motivation**
>
> We thank the reviewers for raising the issue regarding the clarity of the Platonic Representation Hypothesis (PRH) and its connection to the subsequent methodology. We appreciate the opportunity to clarify. We acknowledge that the original presentation of the beginning part of our methodology may have been under-explained, which could lead to misunderstanding. We have updated Section 2.1 (renamed as Motivation from Platonic Representation Hypothesis) with more clarity in the revised version. Below, we provide a response-wise explanation of the relationship between PRH, the generalized representation equation, and the subsequent focus on data perturbation evaluation.
>
> In our work, PRH serves primarily as a **conceptual motivation**, rather than a formal theoretical assumption or theorem. We do not aim to prove PRH, nor is our paper intended as a theoretical contribution. Instead, it reflects the empirical observation that representations tend to converge in practice across various domains.
>
> Then the Equation (4) comes into picture where it comes from our thought process that we are trying to figure out, at the very high level, **in what form PRH might be expressed in time series**. It is not a theoretical derivation chain, it is just a chain of thought built upon mature literature in fields from nonlinear dynamics, chaos theory, and signal processing. In addition, Equation (4) is not presented as a novel formula derived from theory. Instead, it provides a generalized and intuitive representation that could describe the essence of data augmentation methods widely used in literature, such as Panda, NormWear, and Chronos.
>
> It is important to note that this generalized expression does not directly drive model design. Model architecture is one of the important engineering aspects, serving as the foundation for our core exploration. The expression presents a natural framework to represent varying extents of perturbation across arbitrary numbers of pure time series systems. This leads to the central methodological question: **how to evaluate the extent of perturbation, or in other words, the degree of chaotic behavior in the observed time series system, and to what extent does such evaluation outcome influence the training outcome**. We argue that assessing such a degree of chaos on a given pretrained dataset is crucial because it directly relates to pretraining data quality and, consequently, test-time performance. It is at this stage that the main exploration and methodological contributions of our paper begin.

---

> > ### Author Response · Authors · 2025-11-20
> > **Section 2: Corrected description of ablation studies**
> >
> > We sincerely thank the reviewer for raising concerns about the clarity of the controlled factors used in our ablation study for observing the balance law. This point directly touches on one of our key contributions, and we appreciate the opportunity to clarify it more thoroughly. We apologize that, in the original submission, the experimental results were presented only through figures, without clearly articulating the underlying controlled settings. Importantly, the focus of our study is to understand how **dataset balance**, a measurable distributional characteristic of dynamical systems in a given pretraining dataset, affects the quality of the learned representations. Our ablations are therefore designed to **isolate the role of balance under strictly controlled training conditions**. Figures 5 and 6 in the manuscript presented the results of such controlled ablations, and we have now expanded the descriptions of the experimental setting for better clarity. For completeness, we provide the updated text below, which has also been incorporated into Section 3.2 Ablation Studies in the revised manuscript:
> >
> > **\subsection{Ablation Study: Observation of Balancing Law}**
> >
> > Previous studies have extensively demonstrated the existence of scaling laws in signal modeling. In this work, we posit that the balancing law serves as an accompanying factor that should be considered jointly with scaling. We present the ablation study results illustrating the balancing law. The chaotic-based balance score, as discussed, quantifies the distributional diversity of the pretraining dataset, capturing the balance between different types of dynamical systems, which we hypothesize is crucial for effective generative performance.
> >
> > To examine the direct effect of balance, we construct subsets from the integrated benchmark dataset that share an identical level of data size of 10^5 but differ in their balance scores. Balance scores are computed using weighted Shannon entropy and granularity-based diversity measures derived from clustering the underlying dynamical systems. As shown in Figure 5, models pretrained on datasets with higher balance consistently achieve lower generative error across multiple downstream tasks. These results indicate that balance alone, without changing data size, architecture, and training dynamics, correlates with improved generative behavior.
> >
> > To further strengthen the causal interpretation, we compare two controlled pretrainning settings that manipulate balance scores and data size in opposite directions: a relatively small but balanced dataset versus a relatively large but unbalanced one. The balanced dataset, consisting of 1×10^5 samples with a balance score of 0.73, contrasts with the unbalanced dataset consisting of 2×10^5 samples with a balance score of 0.60. Balance scores are computed using weighted Shannon entropy and granularity measures derived from clustering distributions of different dynamical systems. All model checkpoints share identical architectures, hyperparameters, and pretraining procedures. As shown in the left panel of Figure 6, models pretrained on the smaller but more balanced dataset consistently outperform those trained on the larger, unbalanced dataset across various domains and test settings. This experiment rules out the possibility that the improvement observed in Figure 5 is simply a result of larger or richer datasets. Instead, it highlights the role of balance as an independent and influential dataset property.
> >
> > Finally, to disentangle the relationship between scaling and balance, we conduct a third ablation study replicating the scaling-law pattern across datasets with different balance levels (balance scores = 0.60, 0.73, 0.75). The model architecture is fixed, while data size is varied. As illustrated in the right panel of Figure 6, scaling on more balanced datasets leads to a faster drop in test-time generative error, indicating that balance quality accelerates the benefit of scaling. For clarity of presentation, we did not include the results from the physiological signal subset in Figure 6, as its error scale is substantially higher than the other datasets. Nonetheless, these results, reported in the Appendix J, are consistent with and further reinforce the observed trend.

---

> > > ### Author Response · Authors · 2025-11-20
> > > **Section 3: Ablation Study: Alternative Backbones**
> > >
> > > We appreciate the reviewers’ suggestion to examine additional model architectures, both to better contextualize the performance of our primary paradigm model and to assess whether the observed balance law consistently appears across different backbone types. We have reproduced the experiments with 2 alternative model architectures: a uni-variate backbone following guidance of PatchTST, and the channel-attention backbone following guidance from Panda (which is also the same approach leveraged by CBraMod). We report the overall performance below, and these results are also incorporated in the revised version. We’ve also replicated the core ablation study for observing balance law using the checkpoints from univariate backbone. We hope that these additional experimental results, together with the updated description of the ablation study setup, provide clearer context for the proposed framework and reinforce the consistency of our observations.
> > >
> > > ## Section 3.1. Model Pretraining with Varied Architecture
> > > ### Metric: MAE
> > > | Task    | Chronos_base       | Uni-variate Backbone | [CLS] attn. Backbone | Channel attn. Backbone |
> > > |-----------------------------|-----------------|--------------------|--------------------|----------------------|
> > > | Short term forecast | 0.433 ± 0.018   | 0.564 ± 0.019| 0.558 ± 0.010| 0.653 ± 0.032  |
> > > | Long term forecast   | 0.638 ± 0.083   | 0.694 ± 0.022 | 0.696 ± 0.019 | 0.747 ± 0.030  |
> > > | Short term simulate  | 0.451 ± 0.018   | 0.569 ± 0.012  | 0.596 ± 0.011 | 0.405 ± 0.053  |
> > > | Long term simulate  | 0.696 ± 0.135   | 0.706 ± 0.019  | 0.702 ± 0.018 | 0.477 ± 0.062   |
> > > | All short term generative  | 0.442 ± 0.017   | 0.566 ± 0.015  | 0.577 ± 0.010  | 0.529 ± 0.057   |
> > > | All long term generative  | 0.667 ± 0.105   | 0.700 ± 0.020   | 0.699 ± 0.017   | 0.612 ± 0.063        |
> > > | All generative series    | 0.554 ± 0.072   | 0.633 ± 0.021      | 0.638 ± 0.017   | 0.571 ± 0.060        |
> > >
> > > ### Metric: R²
> > > | Dataset   | Chronos_base       | Uni-variate Backbone | [CLS] attn. Backbone | Channel attn. Backbone |
> > > |-----------------------------|-----------------|--------------------|--------------------|----------------------|
> > > | Battery Li-ion SOH    | 0.036 ± 0.026   | -0.190 ± 0.059 | 0.229 ± 0.021   | -0.487 ± 0.075   |
> > > | Battery Na-ion SOH  | 0.966 ± 0.000   | 0.525 ± 0.255   | 0.963 ± 0.0002   | -1.628 ± 3.145   |
> > > | Battery Zn-coin SOH     | 0.433 ± 0.017   | 0.630 ± 0.016    | 0.563 ± 0.012      | 0.541 ± 0.054        |
> > > | Battery SOH Avg   | 0.507 ± 0.173   | 0.297 ± 0.189      | 0.531 ± 0.238    | -0.955 ± 4.986   |
> > >
> > > ### Metric: AUC-ROC
> > > | Task  | Chronos_base       | Uni-variate Backbone | [CLS] attn. Backbone | Channel attn. Backbone |
> > > |-----------------------------|-----------------|--------------------|--------------------|----------------------|
> > > | Wearable State Recognition   | 0.799 ± 0.11    | 0.804 ± 0.25       | 0.813 ± 0.021      | 0.814 ± 0.023        |
> > > | Wearable EEG Tasks           | 0.807 ± 0.021   | 0.875 ± 0.027      | 0.872 ± 0.027      | 0.872 ± 0.026        |
> > > | Wearable Vital Sign (1-MAPE) | 0.953 ± 0.001   | 0.951 ± 0.001      | 0.953 ± 0.001      | 0.949 ± 0.001        |
> > > | Wearable Disease Risk        | 0.621 ± 0.034   | 0.743 ± 0.037      | 0.759 ± 0.034      | 0.724 ± 0.045        |
> > > | Wearable Avg.                | 0.768 ± 0.032   | 0.832 ± 0.028      | 0.838 ± 0.025      | 0.826 ± 0.031        |
> > > | Average Rank Score | 2.250 | 2.688 | **2.188**| 2.750 |
> > >
> > > ## Section 3.2. Reproducing controlled ablation study using univariate backbone
> > > ### Observation of balance law in general.
> > > | Pretrain Subset (Avg. balance) | Short Term Forecast | Long Term Forecast | Short Term Simulate | Long Term Simulate |
> > > |--------------------------------|------------------|-----------------|------------------|-----------------|
> > > | 0.891  | 0.768 ± 0.030    | 0.834 ± 0.031   | 0.767 ± 0.027    | 0.831 ± 0.031   |
> > > | 1.003     | 0.678 ± 0.030    | 0.764 ± 0.031   | 0.690 ± 0.026    | 0.772 ± 0.026   |
> > > | 1.139   | **0.564 ± 0.019**    | **0.694 ± 0.022**   | **0.569 ± 0.012**    | **0.706 ± 0.019**   |
> > >
> > > ### Observation of balance law accompanies scaling law.
> > > | Avg. balance | Data size 10³ | Data size 10⁴ | Data size 10⁵ |
> > > |--------------|---------------|---------------|---------------|
> > > | 1.003        | **0.749 ± 0.029** | 0.743 ± 0.032 | 0.726 ± 0.028 |
> > > | 1.139        | 0.778 ± 0.028 | **0.741 ± 0.034** | **0.633 ± 0.021** |

---

### Author Response · Authors · 2025-11-25
**Summary of Response and Revision**

Dear Reviewers:

We sincerely appreciate the reviewers’ insightful comments and valuable suggestions. We have carefully addressed all points raised and revised the manuscript accordingly, with the updated sections marked in blue. To assist your evaluation, we summarize below the core comments together with our responses and the corresponding revisions.


## Questions, Concerns, and Revisions:
***Issue 1).*** Reviewer **ho8Y**, **nJHj**, **qH8D**, and **ArMM** raise concerns on the unclarity of the role of Platonic Representation Hypothesis (PRH) that has been discussed at the beginning of the method section.

- **Revision:**
    - We have replaced the PRH section from *Method** section to an individual new section: **Section 2. Motivation from Platonic Representation Hypothesis**.
    - Visual components in **Figure 1** is re-arranged with an updated flow of elements to accompany demonstration of the motivation from PRH in the time series domain.

***Issue 2).***  Reviewer **ho8Y** and **qH8D** raises questions around the pattern of influence from factors including data scale and the balancing score. Reviewer **nJHJj** and **ArMM** raises concerns around the same topic that it is not clear where exactly the general performance benefit comes from and to what extent does balancing notion influence the framework behavior.

- **Revision:**
    - We apologize that we realized that we didn’t deliver the detailed experimental settings and present only the result figures of this core ablation studies with varied factors being controlled. For better clarity, we have included a detailed description of factors being controlled during the ablation study, mainly focused around data size, balance score, training setting, and model size. These content are updated in **Section 4.2. Ablation Studies: Observation of Balancing Law**.
    - To further strengthen the observation following suggestions from Reviewer **nJHj** and **ArMM**, we reproduce the ablation studies with alternative backbones where the results are updated both in **Figure 5** and **Appendix J. Table 10 and 11**.

***Issue 3).***  Reviewer **ho8Y** and **ArMM** indicating that the report of the variability in performance is missing.

- **Revision:** We have updated the reported results with mean and performance variability across varied task scenarios within each sub-domains during the evaluation, both in our response and the main tables in **Section 4. Experimental Results**.

***Issue 4).***  Reviewer **nJHj** and **qH8D** both indicating that some of the methodology in the proposed framework are presented with insufficient details, including chaotic-metrics, clustering, and data flow in the paradigm model.

- **Revision:**
    - Procedure of the computation of chaotic metric is provided in updated **Appendix A** with detailed pseudo-code.
    - Detailed formulas describing the data flow is updated in **Appendix D**.

***Issue 5).*** Reviewer **qH8D** indicating that the paper presents limited reflection on the limitation aspect of the work. Reviewer **ho8Y** provide suggestions for discussing the potential boundary of PRH in the time series domain. Reviewer **nJHj** suggest to discuss the tradeoffs between general and domain-specific performance balance. Reviewer **ArMM** suggest us to elaborate the advantage of chaos balance apart from improved accuracy.

- **Revision:** We apologize for including a wrong limitation section in the original version. We have updated the discussion and limitation section incorporating all the suggestions from the reviewers in **Section 5. Conclusion and Discussion**.

---

## Other concerns
In response to reviewer ho8Y, we incorporated an extended **horizon error shift analysis** to further clarify the model’s temporal behavior. For reviewer ArMM, who raised concerns regarding the significance of the reported improvements, we highlighted the exact location of the statistical tests in the manuscript and provided test results demonstrating that the performance gains are **statistically significant**. Regarding reviewer nJHj’s concern that our baselines were not sufficiently solid, we first clarified that the models we compared against are in fact **the most recent and state-of-the-art (SoTA) methods from this year**; moreover, following the suggestion for alternative backbones, **we added experiments using both an up-to-date univariate backbone and a SoTA multivariate backbone, and we reproduced the full set of balancing-law ablation studies under these architectures**. This extensive set of additional experiments further strengthens our conclusions and simultaneously addresses the alternative-backbone suggestions raised by both reviewers ho8Y and ArMM.

We once again thank all reviewers for their time and thoughtful comments. We have carefully considered and addressed all issues raised, and we hope that the revised version adequately addresses all your concerns.

Best regards,

Authors

---

### Author Response · Authors · 2025-11-28
**Letter to Area Chairs: Clarification of Rebuttal and Addressed Issues**

Dear Area Chairs,

We would like to express our sincere appreciation for your tremendous effort during this challenging period. To best support your evaluation, we provide below a concise overview of our revisions and the factual review status of our paper prior to the system rollback.

Firstly, we are grateful for the reviewers’ overall recognition of our contributions. In particular, they highlighted several key strengths of our work:
- **Chaos-theory–based evaluation**: Our work introduces a pioneering approach for assessing the quality of pretraining datasets by employing chaos-theory metrics to quantify the diversity and dynamical coverage of time-series data, offering a principled framework for dataset evaluation in this domain (ho8Y, nJHj, qH8D, ArMM).
- **Cross-domain applicability**: Both the pretraining and evaluation exhibit robust cross-domain generalization, underscoring the modality-agnostic nature of our approach and its suitability for a broad range of scientific, industrial, and healthcare time-series applications (ho8Y, qH8D).
- **Balance law insight** The work proposes a balance law as a companion dimension to dataset scaling, highlighting that qualitative dataset balance can be as important as dataset size. (nJHj, qH8D)
- **Promising empirical performance**: The presented paradigm pretrained foundation model, PANGU-TS, demonstrates leading performance across generative and downstream tasks, outperforming existing baselines in multiple settings. (qH8D, ArMM)

---

We also greatly appreciate the constructive feedback from the reviewers, which has substantially strengthened our work. We are grateful that the reviews were of high quality and converged on three major issues that were raised or noted by all four reviewers:
- The role of the Platonic Representation Hypothesis and its connection to the overall methodology.
- The clarity of how the balance law is observed and whether the observation holds across controlled ablation studies.
- Whether the balance law remains consistent when pretraining with different model architectures.

To directly resolve these common concerns, we have provided detailed clarification and revisions to the manuscript, and conducted extensive additional experiments. The full responses are provided in the section *Response to Common Concerns*.

Following our rebuttal, **one reviewer raised their recommendation to an accept-level score on November 19th, eight days before the security incident**, while three reviewers did not have an opportunity to update their recommendations before the discussion was locked. The reviewer who updated their score confirmed that all of their concerns, **including the three common concerns** mentioned above, had been fully addressed, writing:

> Thank you for the opportunity to review the authors’ revised manuscript and detailed responses. After carefully examining the updated version, I believe that the authors have sufficiently addressed the key concerns raised in the original review... Overall, the revisions improve the clarity, methodological transparency, and contextual framing of the work. I have raised the overall score.

---

In addition to addressing the common concerns in our rebuttal, we provided point-by-point responses to all remaining reviewer comments. Our responses, including results from extensive new experiments, are summarized in *Summary of Response and Revision*. Once again, we sincerely appreciate your valuable contributions to the review process, and we hope that this documented history of the review progression can be taken into account to assist your final assessment.

Best Regards,

The Authors

---

### Meta-Review · Area_Chair_HHqy · 2026-01-05

**Summary:**

The paper received mixed to negative evaluations, with only one reviewer moving to a positive score and doing so with low confidence. While the authors added clarifications, revised figures, and additional experiments, several core concerns remain unresolved, particularly regarding the conceptual connection to the Platonic Representation Hypothesis (PRH), the lack of theoretical justification for the proposed methodology, and incomplete or indirect responses to reviewer questions. Multiple reviewers remain unconvinced that the proposed framework offers a clear methodological or empirical advancement over existing approaches, or that the balance law is sufficiently grounded beyond heuristic observations. The expected final reviewer scores average to approximately 3.5, with limited confidence on the positive side, and the overall assessment does not support acceptance.

**Reviewer Concerns:**

Reviewer ho8Y

- Concerns that are potentially addressed:

   - Statistical reporting and variability: The authors updated results to include mean performance and variability across tasks, which partially addresses the concern on robustness.

    - [Partially resolved] Controlled ablation studies: The authors provided a corrected and more detailed description of their ablation study setup and controlled factors. However, it remains unclear how the specific condition of holding total tokens, sequence length, and number of variates constant was handled, which was explicitly raised by the reviewer.

- Concerns that might not be addressed:

     - Limited theoretical grounding for the Platonic Representation Hypothesis: The authors clarify that PRH is intended only as conceptual motivation. After reviewing the revised manuscript and responses, the connection between PRH and the proposed methodology remains heuristic and not convincingly tied to the core contributions.

     - Reliance on MSE reconstruction loss only: The authors acknowledge this limitation and state that physics or spectral consistency is not the focus of the work. This response does not directly address the reviewer’s concern and is unlikely to change their evaluation.

     - Analysis of error accumulation in long rollouts: While the authors added comparisons between short and long horizon forecasting, the analysis remains coarse and does not fully address concerns about error drift, instability, or noise amplification.

     - Alternative backbones and representation convergence: The added experiments with alternative backbones are informative, but they do not directly answer the reviewer's question regarding whether learned representations converge to a common latent space across fundamentally different architectures such as CNN-based or state space models.


Reviewer nJHj

- Concerns that are potentially addressed:

     - Choice of baselines and comparison with NormWear: The authors provided justification for baseline selection and clarified the use of NormWear for specific downstream tasks.

     - Diagram clarity: The revised figures improve presentation and readability.

- Concerns that might not be addressed:

    - Connection to the Platonic Representation Hypothesis: Similar to Reviewer ho8Y’s concern, the link between PRH and the proposed framework remains insufficiently justified.

     - Theoretical justification of the methodology: While references to chaos theory are provided, they do not establish a clear theoretical foundation explaining why the proposed balance metrics should lead to improved representation learning.

     - Reduced performance on chaotic datasets: The authors’ response does not clearly resolve concerns regarding degraded performance in certain chaotic settings.

     - Ablation on the mixing scheme: The reviewer explicitly requested an ablation study on the effectiveness of the specific mixing scheme, which does not appear to be directly addressed.

Reviewer qH8D

- Concerns that are potentially addressed:

     - The reviewer indicated that concerns related to PRH explanation, balance versus scale, implementation details, and privacy risks were adequately addressed, and increased the score to 6.


Reviewer ArMM

- Concerns that are potentially addressed:

     - Random seed sensitivity: The authors state that varying random seeds does not materially affect results, which partially addresses this concern.

- Concerns that might not be addressed:

     - Whether the proposed foundation model architecture clearly outperforms existing approaches: The additional ablation studies do not directly resolve this concern.

     - Positioning of the paper: It remains unclear whether the paper is intended primarily as an analysis of dataset balance or as a proposal of a new foundation model.

     - Use of unpublished baselines: While the authors responded that the baseline is gaining traction in the community, the concern remains debatable and unresolved from the reviewer’s perspective.


Common concerns across reviewers:
- The connection between the Platonic Representation Hypothesis and the proposed methodology remains heuristic and insufficiently justified, with no clear theoretical or methodological linkage.

- The paper lacks strong theoretical grounding explaining why chaos-theoretic metrics are appropriate or necessary.

- The positioning of the paper is unclear, particularly whether it is intended primarily as an analytical study or as a proposal for a new foundation model.

Additional AC comment:
- The paper would benefit from additional work to be more self-contained, as several results are presented only in tabular form without sufficient explanation, and some important concepts, including core ideas, are not adequately explained in the manuscript.

**Reviewer Scores:**

Reviewer ho8Y: 2 with confidence 3

Reviewer nJHj: 4 with confidence 3

Reviewer qH8D: 6 with confidence 2 (from 4 with confidence 2)

Reviewer ArMM: 2 with confidence 3

Given the halted discussion and the nature of the remaining concerns, further score updates appear unlikely. Even with the positive assessment from Reviewer qH8D, the low confidence and lack of consensus among reviewers suggest limited support.

The expected final average rating is approximately 3.5.

---

### Decision · Program_Chairs · 2026-01-26

Reject